# DISCOVERING LOGIC-INFORMED INTRINSIC REWARDS TO EXPLAIN HUMAN POLICIES

## ABSTRACT

In high-stakes fields like healthcare, it is crucial to distill valuable strategic insights from expert clinicians. This paper focuses on extracting these knowledge-based insights from demonstrations provided by experts, where we represent the knowledge as a set of logical rules. Our learning framework is built upon the classic Inverse Reinforcement Learning (IRL). We assume that experts, like clinicians, are rational, and the treatments they choose are the best choices based on their logical understanding of the situation. Our algorithm can automatically extract these logical rules from their demonstrations. We introduce a neural logic tree generator, which is trained to generate logical statements step by step, starting from the goal and working backward. This mirrors the way humans engage in backward reasoning. Similarly, we interpret policy planning as a forward reasoning process, where the optimal policy is determined by finding the best path forward based on the provided rules. The neural logic tree generator and the policy are learned using the IRL until convergence. This process ultimately leads to the discovery of the most effective strategic rules. As a bonus, our algorithm also allows us to recover the reward function. In our experiments, we demonstrate that our method excels at discovering meaningful logical rules, particularly in the context of healthcare.

## 1 INTRODUCTION

Although deep reinforcement learning has been used in *high-stakes* systems such as healthcare to aid clinicians in making medical decisions (Komorowski et al., 2018), the *lack of interpretability* of the learned *black-box* policies hinders their wide applications in real life. Clinicians are not satisfied with knowing which action to take in certain situations but are also interested in understanding why to take such actions.

In this paper, we focus on answering the following question: *how to automatically uncover the intrinsic logical knowledge to guide the reward design and explain the observational state-action trajectories from expert demonstrations?* In clinical decision systems, it is desirable to extract descriptive and high-level explanatory rules based on disease phenotypes and therapies from demonstrations by clinicians. These discovered logical rules can enhance the sharing of clinical insights, improve the interpretability of medical policies, and contribute to the refinement of treatment strategies.

*Reward engineering* has been a longstanding barrier in many complex decision-making problems. This paper aims to reveal *logic-informed* reward functions and policies by modifying the classic IRL framework (Ng et al., 2000; Finn et al., 2016b; Fu et al., 2018). The policies, reward functions, and their explanatory logical rules will be automatically learned from the data. Notably, our IRL framework automatically mines logic rules from data, which may reduce the cognitive bias introduced by humans and alleviate the human workload. Moreover, the discovered logic rules can be transferred to similar tasks to assist in reward design and policy learning, mitigating the sample efficiency bottleneck and addressing the sparse reward challenge.

Our overall learning framework is built upon the classic *inverse reinforcement learning* (IRL). Our IRL involves two stages: logic tree learning and policy learning. To facilitate efficient and differentiable rule discovery, we introduce a *neural logic tree generator* that generates a composition of logic variables in a sequential manner by mirroring the *backward* reasoning process. It starts with generating the goal and then subsequently generating the symbolic conditions necessary to achieve that goal. Our formulated IRL will drive the logic tree generator to adeptly capture the probabilistic distributions of the most effective strategic rules.

Our policy learning, however, can be regarded as forward reasoning, where the agent is optimized by identifying the most favorable path for forward chaining, given the discovered rules. The proposed IRL algorithm *alternates between neural logic tree learning and policy learning until convergence, resembling a cycle of backward and forward reasoning*. Once converged, the algorithm yields the best set of explanatory logic rules and learned policies. As a by-product, the reward function can be recovered.

To address reward identification challenges in stochastic environments (Ng et al., 2000), we employ the Deep-PQR algorithm (Geng et al., 2020b), a recent IRL method. Deep-PQR introduces the concept of "anchor actions" to facilitate the unique recovery of true reward functions. Overall, our IRL algorithm strategically optimizes the logic generator and policy function until convergence, subsequently recovering the Q-function and reward function. Ultimately, the resulting logic tree informs reward engineering and explains expert policies.

**Our contributions can be summarized as follows**:

(*i*) We present a novel IRL framework that simultaneously learns experts' logical reasoning processes and policies from observational data, enhancing interpretability compared to black-box solutions.

(*ii*) We introduce a neural rule generator capable of incrementally expanding symbolic trees using arithmetic and logical operators to explain experts' demonstrations. This expansion process is learned in a differentiable way.

(*iii*) Our reward learning framework is both tractable and efficient, allowing us to learn intrinsic and unique rewards guided by symbolic logic trees.

(*iv*) We evaluate our method on various datasets, including synthetic and a real-world healthcare scenario. The experimental results demonstrate that our symbolic reward learning framework can outperform most state-of-the-art methods.

## 2   PRELIMINARIES: EXPERT BEHAVIOR MODELS

We consider an *agent behavior model* under the assumption of maximizing the *entropy-augmented* long-term expected reward. The Markov decision process (MDP) is represented by $(\mathcal{S}, \mathcal{A}, \mathcal{P}, r, \rho_0, \gamma)$, where $\mathcal{S}$ denotes the state-space, $\mathcal{A}$ denotes the action-space, $\mathcal{P}$ represents the transition probability distribution, $r(\mathbf{s}, \mathbf{a})$ corresponds to the reward function, $\rho_0$ is the initial state distribution, and $\gamma \in (0, 1)$ is the discount factor. State and action variables at time $t$ are denoted as $\mathbf{s}_t$ and $\mathbf{a}_t$, respectively. $\pi(\mathbf{s}, \mathbf{a})$ represents the stochastic policy of agents, indicating the conditional probability $p(\mathbf{a}_t | \mathbf{s}_t)$. We model the agent's behavior $\pi$ as follows:

$$\max_{\pi} \sum_{t=0}^{\infty} \gamma^t \mathbb{E}\left[ r\left(\mathbf{s}_t, \mathbf{a}_t\right) + \alpha \mathcal{H}\left(\pi\left(\mathbf{s}_t, \cdot\right)\right) \mid \mathbf{s}_0 = \mathbf{s} \right], \tag{1}$$

where $\mathcal{H}(\pi(\mathbf{s}, \cdot)) := -\int_{\mathcal{A}} \log(\pi(\mathbf{s}, \mathbf{a}))\pi(\mathbf{s}, \mathbf{a}) d\mathbf{a}$ is the information entropy. $\alpha \mathcal{H}\left(\pi\left(\cdot \mid \mathbf{S}_t\right)\right)$ with $\alpha > 0$ is introduced to encourage the exploration of the agent behavior $\pi(\mathbf{s}, \mathbf{a})$. If we assume the stochastic policy takes a Boltzmann distribution, also known as stochastic energy-based policy, i.e., $\pi(\mathbf{s}, \mathbf{a}) = \frac{\exp(-\mathcal{E}(\mathbf{s}, \mathbf{a}))}{\int_{\mathbf{a}' \in \mathcal{A}} \exp(-\mathcal{E}(\mathbf{s}, \mathbf{a}')d\mathbf{a}')}$, where $\mathcal{E} > 0$ is an energy function. One can prove that (Haarnoja et al., 2017) by assuming that agents take energy-based policies, the optimal policy function by solving Eq. (1) will be (see Appendix A for the proof)

$$\pi^*(\mathbf{s}, \mathbf{a}) = \frac{\exp\left(\frac{1}{\alpha} Q(\mathbf{s}, \mathbf{a})\right)}{\int_{\mathbf{a}' \in \mathcal{A}} \exp\left(\frac{1}{\alpha} Q\left(\mathbf{s}, \mathbf{a}'\right)\right) d\mathbf{a}'}, \tag{2}$$

with

$$Q(\mathbf{s}, \mathbf{a}) := r(\mathbf{s}, \mathbf{a}) + \max_{\pi} \mathbb{E}\left\{ \sum_{t=1}^{\infty} \gamma^t \left[ r\left(\mathbf{S}_t, \mathbf{A}_t\right) + \alpha \mathcal{H}\left(\pi\left(\mathbf{S}_t, \cdot\right)\right) \right] \mid \mathbf{s}, \mathbf{a} \right\}. \tag{3}$$

Moreover, the likelihood of the observed demonstrations $\tau = \{\mathbf{s}_0, \mathbf{a}_0, \mathbf{s}_1, \mathbf{a}_1, \cdots, \mathbf{s}_T, \mathbf{a}_T\}$ can be derived (using the chain rule) as

$$p(\tau) = \prod_{t=0}^{T} \pi^*\left(\mathbf{s}_t, \mathbf{a}_t\right) \prod_{t=0}^{T-1} p\left(\mathbf{s}_{t+1} \mid \mathbf{s}_t, \mathbf{a}_t\right). \tag{4}$$

Eq. (2) shows that the optimal agent behavior will take the actions that yield higher accumulative reward with exponentially higher probabilities. In our paper, we assume that *the experts are adopting the optimal policy of form Eq. (2)*.

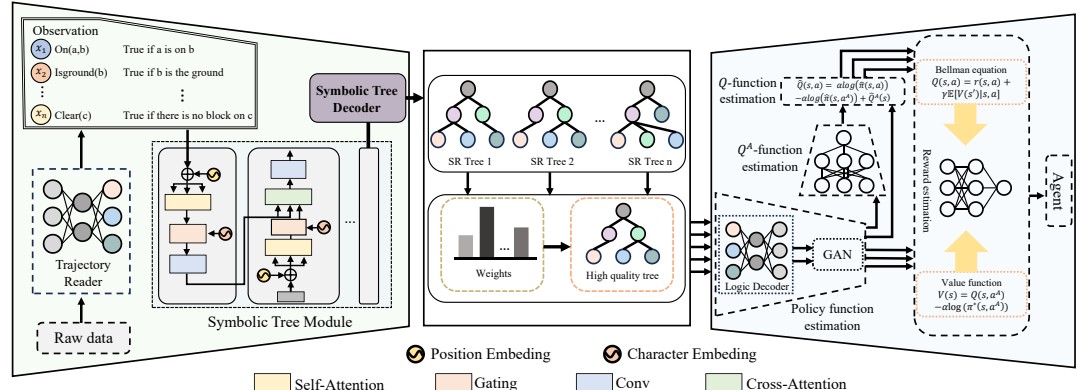

Figure 1: Overview of our proposed framework. Left segment: The diagram depicts the architecture of the neural logic tree. We utilize a Transformer-based autoregressive generator to create the logic trees in a sequential manner. Right segment: The diagram illustrates our comprehensive logic-informed Inverse Reinforcement Learning (IRL) framework. The IRL framework employs a Generative Adversarial Network (GAN)-based cost learning approach, where we formulate a minimax problem to simultaneously train the neural logic generator and the policy. Additionally, the reward function can be uniquely recovered as a by-product of this process.

## 3 LOGIC TREE GENERATOR

Our key idea involves the introduction of a *neural logic tree generator*, from which *a collection of logic rules that need to be executed to reach the goal* will be generated in a top-down fashion, mirroring the *backward* reasoning process of humans. As shown in the left segment of Fig. 1, our generator takes the observed state-action demonstrations as input, and initially encodes them into the *symbolic predicate space*. Subsequently, it utilizes a Transformer-based autoregressive generative model to produce the logic trees. The obtained logic trees serve as symbolic explanations for the expert demonstrations. Moreover, they will be subsequently leveraged in the policy and reward learning phase. We will defer to Section 4 to discuss the IRL algorithm in terms of policy and reward learning, where the parameters of our logic tree generator will also be learned.

### 3.1 SYMBOLIC LOGIC TREE

Let $\mathcal{X}$ be a predicate set $\mathcal{X} = \{X_1, X_2, \ldots, X_n\}$, where each predicate $X_i \in \{0, 1\}$ is Boolean logic variable. Formally, a symbolic tree $\mathcal{R}$ consists of a set of logic rules:

$$\bigcup_{k \in \mathcal{K}} \left\{ f_k \,\middle|\, f_k := \bigwedge_{u \in I_k^1} X_u \wedge \bigwedge_{v \in I_k^0} \neg X_v \right\}, \quad (5)$$

where each logic rule $f_k$ from the rule index set $\mathcal{K}$ is written as a conjunctive clause, and $I_k^1$ (resp. $I_k^0$) is the index set of the variables in $\mathcal{X}$ without (resp. with) a NOT operator. An example of the logic tree $\mathcal{R}$ considered in our paper is shown in Fig. 2, where the tree defines a collection of logic rules, indicating the conditions that need to be satisfied to reach the goal.

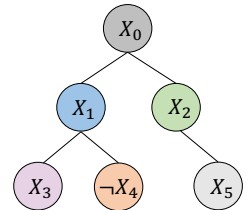

Figure 2: Illustration of a logic tree $\mathcal{R}$. The tree indicates that to reach the goal $X_0$, the following logic rules: $X_0 \leftarrow X_1 \wedge X_2$, $X_1 \leftarrow X_3 \wedge \neg X_4$, and $X_2 \leftarrow X_5$ must be executed.

### 3.2 AUTOREGRESSIVE GENERATIVE MODEL

Our (amortized) neural logic tree generator is designed to generate a symbolic logic tree with a distribution $p_\phi(\mathcal{R} \mid \tau)$, conditional on a state-action trajectory $\tau$. Specifically, we generate a symbolic logic tree $\mathcal{R}$ as a *sequence* in an autoregressive manner, according to a *pre-order traversal*, i.e.,

$$p_\phi(\mathcal{R} \mid \tau) = p_\phi(X^g \mid \tau) \prod_{l=2}^{L} p_\phi(X^l \mid \tau, \boldsymbol{X}^{<l}), \quad (6)$$

where the index $l$ refers to the position in this pre-order traversal, and $X^0 = X^g$ is the goal predicate, and $\boldsymbol{X}^{<l} = (X^0, \ldots, X^{l-1})$. Our logic generator starts from the root node, representing the final goal, and generates subsequent subgoals and conditions.

Our symbolic tree generator contains three main components: (1) a *trajectory reader* encodes the raw trajectory data; (2) an *abstract symbolic tree reader* encodes the predicates and the previously generated partial symbolic tree; (3) a *decoder* integrates the node scheduled for expansion along with inputs from the two previous readers to predict the next predicate.

To construct $p_\phi(\mathcal{R} \mid \tau)$, we begin a trajectory reader, which tokenizes $\tau$ into the predicate space, resulting in a sequence of ordering $X_{(1)}, X_{(2)}, \ldots, X_{(n)}$, where $n$ denotes the length of the grounded predicate sequence. This tokenization is performed using a set of predefined labeling functions. Then each grounded predicate $X_{(i)}$ is further divided into characters $c_1^{X_{(i)}}, c_2^{X_{(i)}}, \ldots, c_m^{X_{(i)}}$ by the first block of the abstract symbolic tree reader, where $m$ represents the number of characters in $X_{(i)}$. The character here represents the state/action which is associated with this predicate. All predicates and characters are embedded as real-valued vector $\mathbf{X}_{(1)}, \mathbf{X}_{(2)}, \ldots, \mathbf{X}_{(n)}$ and $\mathbf{c}_1^{X_{(i)}}, \mathbf{c}_2^{X_{(i)}}, \ldots, \mathbf{c}_m^{X_{(i)}}$, then we can represent a predicate by character embeddings with a fully-connected layer. The first block of the abstract symbolic tree reader contains three different sub-layers: self-attention, gating mechanism, and convolution, designed to extract features. The neural structure of our reader follows the Transformer architecture and uses multi-head attention to capture long-dependency information.

While our trees are generated by predicting sequences of predicates, these predicates alone lack a concrete representation of the tree's structure, making them insufficient for predicting the next predicate. Therefore, the rest blocks of the abstract symbolic tree reader were used to incorporate both the predicted predicates and the tree's structural information. Specifically, We encode the rules using an attention mechanism and subsequently employ a tree convolution layer to amalgamate the encoded representation of each node with its ancestors. Our final component is a decoder that integrates information from generated logic rules with the state-action trajectory description and predicts the next predicate. The decoder treats the non-terminal node to be expanded as a query, represented as a path from the root to the node to be expanded. For detailed input descriptions and neural structures for each module, please refer to Appendix B.

## 4 LEARNING FRAMEWORK

Next, we introduce our innovative logic-informed IRL framework, which aims to learn three key components: the logic tree generator represented as $p_\phi$, the expert policy, and the reward function. Our algorithm operates by iteratively updating the expert policy and the rule generator $p_\phi$ until convergence. Subsequently, we recover the Q-function and reward function in a specific order. We will provide more detailed explanations in the following sections.

### 4.1 JOINT ESTIMATION OF EXPERT POLICY AND LOGIC GENERATOR

We first adopt a GAN-like adversarial IRL framework (Finn et al., 2016b) to *jointly* estimate the expert policy and the logic tree generator.

**Logic-Informed Energy Function** Our innovation lies in the design of the *energy function*, which encapsulates *high-level strategic* information revealed from the trajectory, utilizing predefined predicates and logic rules. More precisely, we propose to parametrize the energy function using *logic-informed features* given a generated logic tree (see Eq. (5)), as outlined in the following equation:

$$\mathcal{E}_\theta(\tau; \mathcal{R}) = -\sum_{k \in \mathcal{K}(\mathcal{R})} \theta_k \Big( \sum_{u \in I_k^1(\mathcal{R})} X_u(\tau) - \sum_{v \in I_k^0(\mathcal{R})} X_v(\tau) - |I_k^1(\mathcal{R})| + \epsilon \Big). \tag{7}$$

Here, $0 < \epsilon < 1$, $\theta_k \geq 0$; the index set of logic rules $\mathcal{K}$ and the index set of variables $f_k^1, f_k^0$ are associated with the logic tree $\mathcal{R}$; $|\cdot|$ denotes the cardinality of a set; and all predicate variables $X_u$, $X_v$ are grounded by the trajectory $\tau$. The parameter $\theta_k$ can be designed as a unique parameter for each rule within the tree, or it can be shared among all the rules in the tree. Sharing the parameters across rules can help reduce the number of parameters that need to be learned.

Each summand in Eq. (7) (i.e., the value within the big parentheses), once exponentiated, served as a soft approximation of the logic rule $f_k(\tau)$ as defined in Eq. (5). To illustrate, when $X_u(\tau) = 1$ for every $u \in I_k^1(\mathcal{R})$ and $X_v(\tau) = 0$ for every $v \in I_k^0(\mathcal{R})$, we have that $f_k(\tau) = 1$ and the exponential of the summand equals $e^{+\epsilon}$. Conversely, when $X_u(\tau) = 0$ for every $u \in I_{f_k}^1(\mathcal{R})$ and $X_v(\tau) = 1$ for every $v \in I_k^0(\mathcal{R})$, we have that $f_k(\tau) = 0$ and the exponential of the summand equals $e^{-|I_k^1(\mathcal{R}) \cup I_k^0(\mathcal{R})| + \epsilon}$, which is presumably a very small number that is close to 0. The parameter $\theta_k$

controls the scaling of the exponential as well as the weight of each clause in the logic tree. We will revisit the interpretation of the energy function when we discuss the policy learning.

With a slight abuse of notation, we introduce a logic-informed energy function, which can be interpreted as the *product of experts* (Hinton, 2002), where each generated logic tree sample can be regarded as an expert's logical reasoning used in the energy function.

$$\mathcal{E}_{\theta,\phi}(\tau) = \mathbb{E}_{\mathcal{R} \sim p_\phi(\cdot|\tau)}[\mathcal{E}_\theta(\tau;\mathcal{R})]. \tag{8}$$

Here, each symbolic tree $\mathcal{R}$ independently evaluates the likelihood of a trajectory $\tau$ as $\exp(-\mathcal{E}_\theta(\tau;\mathcal{R}))$. The distribution of expert's logic trees is $p_\phi(\cdot \mid \tau)$ and the expression $\exp(-\mathcal{E}_{\theta,\phi}(\tau))$ represents the combined likelihood.

It is crucial to note that $\mathcal{E}_{\theta,\phi}(\tau)$ is dependent on the learnable parameter $\theta$ and the distribution $p_\phi$ of the logic tree $\mathcal{R}$. During the learning process, the objective is to minimize the energy when applying an expert demonstration $\tau$. In other words, the energy function is trained to increase the likelihood of grounding each predicate $\mathcal{X}_i$ within the symbolic tree $\mathcal{R}$ and the goal predicate. In practice, to make the expectation tractable, we will use the top-$K$ logic trees with generated probabilities to approximate the expectation. In our experiments, we use $K = 1$, and the performance is quite satisfactory.

**GAN Guided Cost Learning**    Now, given the logic-informed energy function, we consider a discriminator of the form

$$\mathcal{D}_{\theta,\phi,\psi}(\tau) = \frac{\exp(-\mathcal{E}_{\theta,\phi}(\tau))}{\exp(-\mathcal{E}_{\theta,\phi}(\tau)) + p_{\pi_\psi}(\tau)}, \tag{9}$$

where $p_{\pi_\psi}$ represents the likelihood of a trajectory $\tau$ induced from a policy $\pi_\psi$, parameterized by $\psi$; $\exp(-\mathcal{E}_{\theta,\phi}(\tau))$ represents the likelihood of a state-action trajectory $\tau$ induced from the logic-informed energy model Eq. (8). The GAN-guided cost learning framework considers a minimax problem

$$\min_\psi \max_{\theta,\phi} \mathbb{E}_{\tau \sim \pi_{\text{expert}}}\big[\log \mathcal{D}_{\theta,\phi,\psi}(\tau)\big] + \mathbb{E}_{\tau \sim \pi_\psi}\big[\log(1 - \mathcal{D}_{\theta,\phi,\psi}(\tau))\big]. \tag{10}$$

The discriminator maximizes the log-likelihood between the encapsulated raw data trajectory and the generated trajectories by the current policy $\pi_\psi$. Practically, we replace $\pi_\psi$ with a mixture between the raw data trajectory and the generated trajectories by the current policy. Given $\psi$, the optimal discriminator satisfies $\mathcal{E}_{\theta^*,\phi^*}(\tau) = -\log p_\psi(\tau)$, at which the output of the discriminator is $1/2$ for all trajectories.

The GAN objective can be trained using gradient-based algorithms with appropriate parameterization. Before that, we can compute the gradient of $-\mathcal{E}_{\theta,\phi}(\tau)$ according to Eq. (8)as follows:

$$\nabla_\phi \mathcal{E}_{\theta,\phi}(\tau) = \mathbb{E}_{\mathcal{R} \sim p_\phi(\cdot|\tau)}[\nabla_\phi \log p_\phi(\mathcal{R} \mid \tau) \cdot \mathcal{E}_\theta(\tau;\mathcal{R})] \tag{11}$$

Therefore, we can compute the gradient of the GAN objective with respect to $\phi$ using the log-derivative trick, which yields:

$$\nabla_\phi \mathbb{E}_{\tau \sim \pi_{\text{expert}}}\big[\log \mathcal{D}_{\theta,\phi,\psi}(\tau)\big] + \mathbb{E}_{\tau \sim \pi_\psi}\big[\log(1 - \mathcal{D}_{\theta,\phi,\psi}(\tau))\big]$$
$$= -\mathbb{E}_{\tau \sim \pi_{\text{expert}}}\big[\nabla_\phi \mathcal{E}_{\theta,\phi}(\tau)\big] + \mathbb{E}_{\tau \sim \pi_{\text{expert}}}\left[\frac{\exp(-\mathbb{E}_{\mathcal{R} \sim p_\phi(\cdot|\tau)}[\mathcal{E}_\theta(\tau;\mathcal{R})])}{\exp(-\mathbb{E}_{\mathcal{R} \sim p_\phi(\cdot|\tau)}[\mathcal{E}_\theta(\tau;\mathcal{R})]) + p_{\pi_\psi}(\tau)} \cdot \nabla_\phi \mathcal{E}_{\theta,\phi}(\tau)\right]$$
$$+ \mathbb{E}_{\tau \sim \pi_\psi}\left[\frac{\exp(-\mathbb{E}_{\mathcal{R} \sim p_\phi(\cdot|\tau)}[\mathcal{E}_\theta(\tau;\mathcal{R})])}{\exp(-\mathbb{E}_{\mathcal{R} \sim p_\phi(\cdot|\tau)}[\mathcal{E}_\theta(\tau;\mathcal{R})]) + p_{\pi_\psi}(\tau)} \cdot \nabla_\phi \mathcal{E}_{\theta,\phi}(\tau)\right]. \tag{12}$$

Policy learning can be regarded as the importance sampling used for estimating the *partition function* of an energy-based model (Finn et al., 2016a). In other words, given the current energy function, the policy is trained to generate state-action trajectories that will yield a low-energy function (i.e., a high probability), resulting in the logic trees being more likely to be satisfied. In essence, policy learning is about finding the most effective path to achieve the end goal given the current logic rules.

Therefore, our overall logic-informed IRL can be seen as a cycle of alternating between backward reasoning and forward reasoning until convergence is achieved. Consequently, we can learn the best probabilistic distribution of the logic trees as well as the policy.

## 4.2 REWARD RECOVERY

After learning the expert policy, we recover the reward function using the deep PQR algorithm (Geng et al., 2020b), which is a modification to the well-known AIRL (Fu et al., 2018) to address the reward ambiguity. This is achieved by first identifying the Q-function under an anchor-action assumption and then estimating the reward function.

**Two-stage Q-function Estimation**  Recall from Eq. (2) the relationship between the expert policy and the optimal Q-function, the ratio remains unchanged if $Q^*(\mathbf{s}, \mathbf{a})$ is shifted by a function $\Psi(\mathbf{s})$. As a result, $Q^*$ is not identifiable in the above form. To resolve this issue, we adopt the anchor-action assumption (Geng et al., 2020a), which postulates the presence of an anchor action $\mathbf{a}^A \in \mathcal{A}$ such that its reward values $r(\cdot, \mathbf{a}^A)$ is fixed beforehand. In our setting, the anchor action is chosen as a "non-action", yielding no reward. We refer to Appendix C for the estimation of the reward values and the Q values associated with the anchor action $\mathbf{a}^A$.

Given the anchor estimator $\hat{Q}^A$ and the policy estimator $\pi_{\hat{\psi}}$, the Q-function is estimated by

$$\hat{Q}(\mathbf{s}, \mathbf{a}) = \alpha \log(\pi_{\hat{\psi}}(\mathbf{s}, \mathbf{a})) - \alpha \log(\pi_{\hat{\psi}}(\mathbf{s}, \mathbf{a}^A)) + \hat{Q}^A(\mathbf{s}).$$

Thereby, we can apply the Fitted-Q-Iteration Identification method, which amounts to solving a simple one-action MDP.

**Reward Estimation**  Given the Q-function estimator $\hat{Q}$ and the policy estimator $\pi_{\hat{\psi}}$, the reward function is estimated in a manner similar to the Bellman equation:

$$\hat{r}(\mathbf{s}, \mathbf{a}) = \hat{Q}(\mathbf{s}, \mathbf{a}) - \gamma \hat{\mathbb{E}}_{\mathbf{s}'}[-\alpha \log(\hat{\pi}(\mathbf{s}', \mathbf{a}^A)) + \hat{Q}(\mathbf{s}', \mathbf{a}^A) \mid \mathbf{s}, \mathbf{a}]. \tag{13}$$

Here, $\hat{\mathbb{E}}_{\mathbf{s}'}$ denotes the estimated expectation over the state variable $\mathbf{s}'$, which represents the one-step look-ahead state variable. The main idea behind the formulation in Eq. (13) is to replace the value function in the Bellman equation with a specification that explicitly incorporates agent policies into the reward functions. This representation avoids the direct calculation of the value function, simplifying the estimation of the expectation. To estimate the expectation, we use a deep neural network given $\hat{\pi}(\mathbf{s}, \mathbf{a})$ and $\hat{Q}(\mathbf{s}, \mathbf{a})$, transforming the reward estimation problem into a supervised learning task.

With this formulation, we can effectively estimate the reward function in a more direct manner without explicitly computing the value function. By considering the impact of agent policies, we obtain a more refined representation of the reward, enabling efficient supervised learning for the reward estimation process. The use of deep neural networks in this approach provides a flexible and scalable framework for reward estimation, making it applicable to a wide range of applications.

We want to emphasize that even though our reward function is defined at the state-action level, its learning process is implicitly influenced by the symbolic logic trees that we've learned. The learned logic trees, being at a higher level of strategy description, are more generalizable than the reward function.

## 5  EXPERIMENT

In this section, to test the robustness of our proposed framework, we conduct several experiments on synthetic and real-world datasets and show that our method can solve a broad set of decision-making tasks, such as block manipulation and healthcare event prediction, and meanwhile provide strategic explanations.

### 5.1  IMPLEMENTAL DETAILS

For all the deep learning-based models, we implement them in PyTorch and train them on Ubuntu 16.04 with 3090 GPU. The batch size is set to 50 for all the methods. The dimension of the final hidden state for prediction is set to 256, i.e., $l = 256$. The layer of RNN or Transformer is set to 1 for all the methods unless there is a hierarchical structure. Dropout methods are used for all the models in the final prediction layer unless there is a default setting. The dropout rate is set to 0.5. Adam (Kingma & Ba, 2014) optimizer is used for all the methods. For the learning rate, we use the grid search approach to select the best one for each method according to the validation set.

### 5.2  SYNTHETIC EXPERIMENTS

We test our model's capability of decision-making in the classic blocks world domain by slightly extending the model to fit the formulation of the Markov Decision Process (MDP) in reinforcement learning. Additionally, we perform experiments specifically designed to validate the accuracy and quality of our estimated reward function.

**Experimental Setting.**  We further show our model's ability to tackle algorithmic tasks, such as `BlocksWorld`, `Sorting`, and `Path`. We view an algorithm as a sequence of primitive actions and

Table 1: Comparison between different methods in the blocks world, sorting integers, and finding shortest paths, where $m$ is the number of blocks in the blocks world environment or the size of the arrays/graphs in the sorting/path environment. The performance is evaluated by two metrics and separated by "/": the probability of completing the task during the test, and the average `Moves` used by the agents when they complete the task.

| Methods | BlocksWorld | | Sorting | | Finding Path | |
|---|---|---|---|---|---|---|
| | m=40 | m=80 | m=40 | m=80 | m=40 | m=80 |
| NLM | 100%/70 | 83%/247 | 100%/41 | 89%/216 | 100%/4 | 100%/32 |
| MemNN | 57%/231 | 0%/N/A | 96%/745 | 74%/1385 | 35%/36 | 0%/N/A |
| MaxEnt-IRL | 42%/264 | 0%/N/A | 82%/811 | 44%/1879 | 26%/49 | 0%/N/A |
| Deep PQR | 78%/125 | 43%/362 | 100%/165 | 70%/553 | 77%/31 | 31%/198 |
| AIRL | 53%/305 | 24%/1147 | 92%/766 | 71%/1406 | 71%/42 | 0%/N/A |
| Ours | 100%/59 | 93%/189 | 100%/41 | 97%/170 | 100%/4 | 100%/29 |

Table 2: Mean Squared Error (MSE) for reward recovery with a different number of blocks, where m is the number of blocks in the blocks world environment.

| Block Num | m=10 | m=20 | m=30 | m=40 | m=50 |
|---|---|---|---|---|---|
| NLM | 0.225 | 0.219 | 0.138 | 0.439 | 0.21 |
| Ours | 0.093 | 0.139 | 0.124 | 0.351 | 0.155 |

cast it as a reinforcement learning problem. Each synthetic dataset contains 1, 000 event streams partitioned into three sets: training (70%), validation (15%), and test (15%). Some information about datasets can be found in Supplementary material.

**Baselines** For IRL, we consider the classic MaxEnt-IRL proposed in Ziebart et al. (2008), which estimates the $Q$-function and treats it as the reward function. Further, we include the AIRL method (Fu et al., 2018), which attempts to distinguish the reward function from the $Q$-function by a disentangling procedure. Moreover, we consider three baselines as representatives of the connectionist and symbolicist: Memory Networks (MemNN) (Sukhbaatar et al., 2015) and Neural Logic Machine (NLM) (Dong et al., 2019). We also make comparisons with other models such as Deep PQR (Geng et al., 2020b).

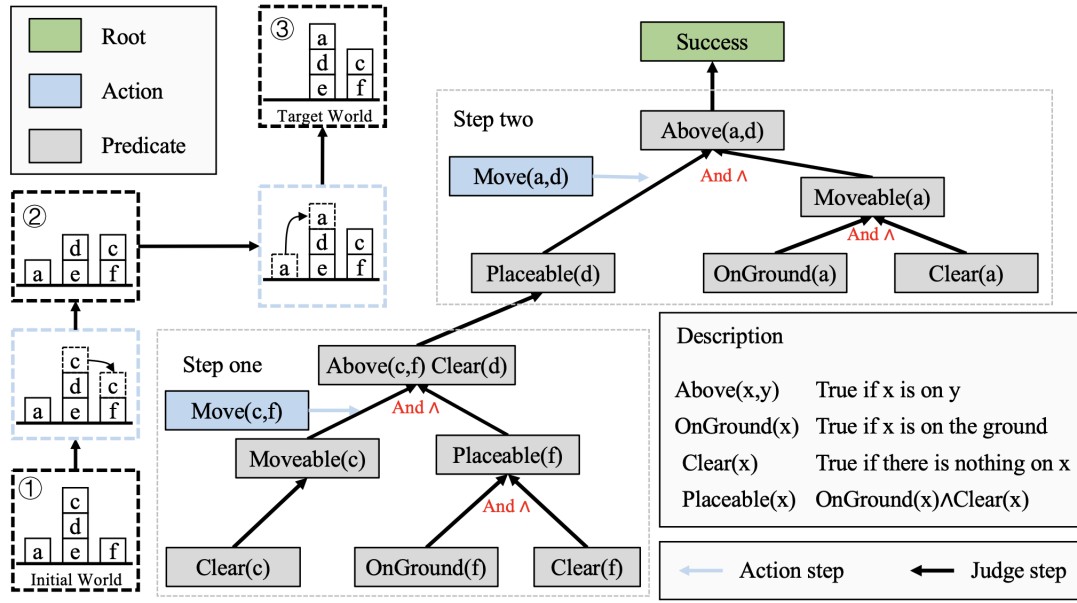

Figure 3: A logic tree induced by our algorithm in BlocksWorld task. In the first step, we move the block c onto the block f. In the second step, we move the block a onto the block d to the target configuration.

**Results.** Our findings, detailed in Tables 1 and 2, demonstrate that our method outperforms others in both performance and reward recovery. Notably, the performance gap between our approach and

Table 3: Diagnosis prediction results on MIMIC-III and MIMIC-IV using w-$F_1$ (%) and R@$k$ (%).

| Methods | MIMIC-III | | | MIMIC-IV | | |
|---|---|---|---|---|---|---|
| | w-$F_1$ | R@10 | R@20 | w-$F_1$ | R@10 | R@20 |
| Chet | 22.63(0.08) | 28.64(0.13) | 37.87(0.09) | 26.35(0.13) | 30.28(0.09) | 38.69(0.15) |
| HiTANet | 21.15(0.19) | 26.02(0.25) | 35.97(0.18) | 24.92(0.27) | 27.45(0.33) | 36.37(0.24) |
| ConCare | 20.94(0.06) | 24.04(0.16) | 34.11(0.12) | 23.59(0.18) | 26.52(0.13) | 35.23(0.07) |
| Deep PQR | 20.86(0.14) | 24.61(0.08) | 34.23(0.11) | 23.13(0.13) | 26.39(0.06) | 35.45(0.19) |
| CGL | 21.92(0.12) | 26.64(0.30) | 36.72(0.15) | 25.41(0.08) | 28.52(0.42) | 37.15(0.29) |
| NLRL | 20.23(0.06) | 24.59(0.09) | 34.03(0.07) | 23.75(0.19) | 26.80(0.14) | 35.70(0.15) |
| Ours | 22.78(0.09) | 29.01(0.11) | 38.10(0.08) | 26.47(0.10) | 30.35(0.07) | 38.65(0.11) |

the Neural Logic Machines (NLM) (Dong et al., 2019) is evident. While both methods achieve close to the best possible result on each task, there is little room for improvement. But ours obtain fewer moves when finishing all tasks. The MemNN (Sukhbaatar et al., 2015) method is ineffective when the size of the arrays/graphs in the sorting/path environment becomes larger. Deep PQR (Geng et al., 2020b) can presumably learn disentangled rewards, but we find that the formulation does not perform well even in learning rewards in the original task, let alone transferring to a new domain. It learns successfully in the training domain but does not acquire a representation that is suitable for transfer to test domains, with a 78% success rate of completing the `BlocksWorld` task (m=40). In contrast, our method not only excels in learning disentangled rewards but also adapts effectively to significant domain shifts, even in high-dimensional environments where exact reward extraction is challenging. The accuracy of our reward function estimates, as evidenced by the results of reward recovery, further validates the alignment of our approach with expert decisions, despite the constraints of our dataset.

Moreover, we also show a most likely logic tree generated from our algorithm in Fig. 3 to demonstrate its effectiveness. Given an initial configuration of blocks world, our goal is to transform it into a target configuration by taking a sequence of Move operations. The learning system should recover a set of rules and generalize to a blocks world that contains more blocks than those encountered during training. In our tree architecture, the leaf nodes define the initial states, and the inner nodes are determined by gating functions encoding the probability of taking the rightmost branch at each leaf node.

## 5.3 REAL-WORLD EXPERIMENTS

**Baselines.** To compare our method with state-of-the-art models, we select the following methods as baselines: 1) RNN/Attention-based model: Chet (Lu et al., 2022), HiTANet (Luo et al., 2020), ConCare (Ma et al., 2020), CGL (Lu et al., 2021), Deep PQR (Geng et al., 2020b) and NLRL (Jiang & Luo, 2019).

**Metrics.** We adopt weighted F1 score (w-$F_1$) and top $k$ recall (R@$k$) for diagnosis predictions. w-$F_1$ is a weighted sum of $F_1$ for each class. R@$k$ is the ratio of true positive numbers in top $K$ predictions by the total number of positive samples, which measures the prediction performance on a subset of classes.

**Results.** We report the mean and standard deviations of evaluation metrics by running each model 5 times with different parameter initializations. Table 3 shows the results of diagnosis prediction using w-$F_1$ (%) and R@k (%). It shows that our method outperforms baselines on both datasets. Note that our method is better than CGL (Lu et al., 2021) even without medical ontology graphs used in these two models. It further validates the significance of learning logic rules in health-care prediction. The attention-based method (Ma et al., 2020) and graph-based method (Lu et al., 2021) can force models to only focus on the visits that contain risk factors and ignore the rest visits. Aggregating all visits together may further induce noise and hurt final performance.

## 5.4 VISUALIZATION OF MIMIC DATASET

To Explain patient trajectories, our real-world example studies decision-making within the MIMIC-III dataset in Fig. 4. Let **Patient A** initially have the sepsis symptoms including above-average body temperature, high respiratory rate, high white blood cell count, and so on, progressing towards rehabilitation, progressing towards rehabilitation. Let **Patient B** is diagnosed with high body temperature

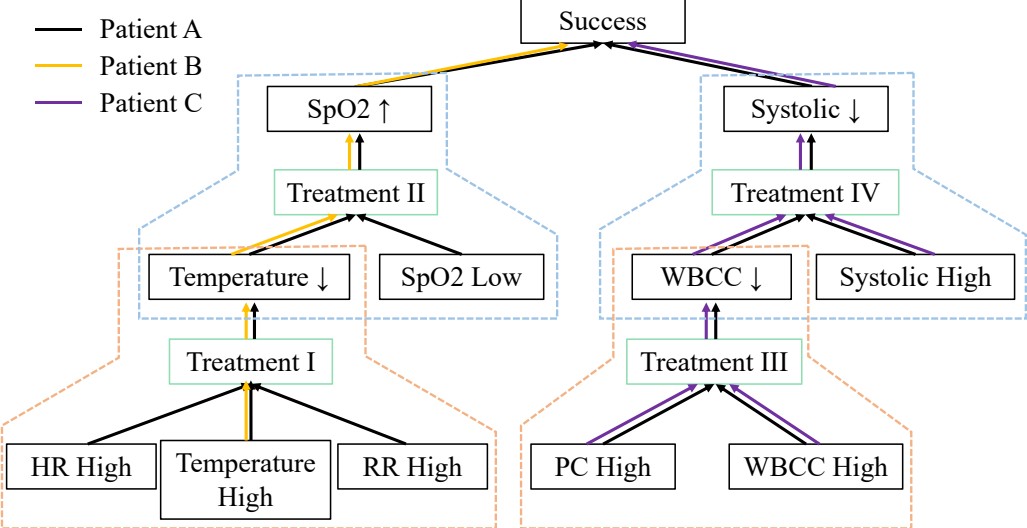

Figure 4: Multidimensional trees of logic strategy generated in MIMIC III dataset. WBCC: White blood cell count; PC: platelets count; RR: Respiratory rate; HR: Heart rate. The leaf nodes present some symptoms of the patient, and each dashed box means that by taking some treatments, the patient may lessen the symptoms. After taking four treatments, the patient fully recovered from all illnesses.

at the first visit without other symptoms, and **Patient C** is diagnosed with high white blood cell count and high platelets count. Take Patient A as an example, our generated strategy gives a certain diagnosis of sepsis by dividing his symptoms into four parts and therapy them separately. Firstly, considering the high temperature, high respiratory rate, and high heart rate, this strategy suggests Treatment I, such as taking the medicine to decrease the body temperature down to the normal level, following treating the low SpO2 by Treatment II at the following time step. Also, the other symptoms (including high platelets count, high white blood cell count and high systolic) are solved by Treatment III and IV hierarchically. Our policy correctly learns that treatments are only needed until the diagnosis is confirmed. Moreover, the symptoms of Patient B and Patient C are the sub-tree of Patient A, so their treatment strategies are also included in this sepsis treatment system, and other complications can also be treated by following this strategy.

## 6 LIMITATIONS AND DISCUSSION

We can enhance our logic-informed IRL framework in several ways: (*i*) Currently, the accuracy of the generated logic trees needs validation by human experts. In future work, we can develop a human-in-the-loop algorithm that allows flexible incorporation of human expert opinions in rule generation. (*ii*) In the current version, users must specify the predicate set in advance. However, can we enable our algorithm to automatically discover new concepts or predicates? (*iii*) At present, we employ a neural logic rule generator. In the future, we may replace it with a different approach like LLMs, and we're interested in evaluating the performance of our logic-informed IRL with this new knowledge generator.

## 7 CONCLUSION

In this paper, we present a logic-informed IRL framework. A neural logic generator is trained to produce logical rules to explain expert demonstrations and their underlying logical reasonings. Using the proposed IRL framework, both the neural logic generator and the policy are learned through a minimax optimization until the algorithm converges. Our overall logic-informed IRL can be seen as a cycle of alternating between backward reasoning and forward reasoning until convergence is achieved. As a bonus, our approach can also uniquely recover the reward functions. We empirically evaluated our methods on synthetic and real healthcare datasets, which demonstrated promising results.

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
