# DISCOVERING LOGIC-INFORMED INTRINSIC REWARDS TO EXPLAIN HUMAN POLICIES - SUPPLEMENTARY MATERIAL

## 1 THE MAXIMUM ENTROPY POLICY

In this appendix, we present proofs for the theorems that demonstrate how a policy function can be monotonically optimized with respect to the maximum entropy objective. Recall the objective function of Max-Entropy RL:

$$\pi^* = \arg\max_\pi \sum_{t=0}^\infty \gamma^t \mathbb{E}_{(\mathbf{s}_t, \mathbf{a}_t) \sim \rho_\pi} \left[ r(\mathbf{s}_t, \mathbf{a}_t) + \alpha \mathcal{H}(\pi(\mathbf{s}_t, \cdot)) \mid \mathbf{s}_0 = \mathbf{s} \right]. \tag{1}$$

This objective corresponds to maximizing the discounted expected reward and entropy for future states originating from every state-action tuple $(\mathbf{s}_t, \mathbf{a}t)$ weighted by its probability $\rho_\pi$ under the current policy. We begin by defining the Q-function for any given policy $\pi$. This Q-value represents the expected total reward, taking into account both rewards and entropy, under the policy $\pi$:

$$Q^\pi(\mathbf{s}, \mathbf{a}) = r_0 + \mathbb{E}\left[ \sum_{t=1}^\infty \gamma^t (r_t + \alpha \mathcal{H}(\pi(\mathbf{s}_t, \cdot))) \right]. \tag{2}$$

The discounted maximum entropy policy objective can now be defined as:

$$J(\pi) = \sum_t \mathbb{E}_{(\mathbf{s}_t, \mathbf{a}_t) \sim \rho_\pi} \left[ Q^\pi(\mathbf{s}_t, \mathbf{a}_t) + \alpha \mathcal{H}(\pi(\mathbf{s}_t, \cdot)) \right]. \tag{3}$$

If we greedily maximizes the sum of entropy and value with one-step look-ahead, then we obtain $\hat{\pi}$ from $\pi$:

$$\mathbb{E}_{\mathbf{a} \sim \pi} \left[ Q^\pi(\mathbf{s}, \mathbf{a}) \right] + \alpha \mathcal{H}(\pi(\mathbf{s}, \cdot)) \leq \mathbb{E}_{\mathbf{a} \sim \hat{\pi}} \left[ Q^\pi(\mathbf{s}, \mathbf{a}) \right] + \alpha \mathcal{H}(\hat{\pi}(\mathbf{s}, \cdot)). \tag{4}$$

When we assume that the entropy parameter $\alpha = 1$, it is worth noting that:

$$\mathcal{H}(\pi(\mathbf{s}, \cdot)) + \mathbb{E}_{\mathbf{a} \sim \pi} \left[ Q^\pi(\mathbf{s}, \mathbf{a}) \right] = -D_{KL} \left[ \pi(\mathbf{s}, \cdot) \, \| \, \hat{\pi}(\mathbf{s}, \cdot) \right] + \log \int \exp(Q^\pi(\mathbf{s}, \mathbf{a})) \, d\mathbf{a}. \tag{5}$$

Then we can show that $Q$ is bounded for any $\mathbf{s}$:

$$\begin{aligned}
Q^\pi(\mathbf{s}, \mathbf{a}) &= \mathbb{E}_{\mathbf{s}_1} \left[ r_0 + \gamma(\mathcal{H}(\pi(\mathbf{s}_1, \cdot)) + \mathbb{E}_{\mathbf{a}_1 \sim \pi} \left[ Q^\pi(\mathbf{s}_1, \mathbf{a}_1) \right]] \right. \\
&\leq \mathbb{E}_{\mathbf{s}_1} \left[ r_0 + \gamma(\mathcal{H}(\hat{\pi}(\mathbf{s}_1, \cdot)) + \mathbb{E}_{\mathbf{a}_1 \sim \hat{\pi}} \left[ Q^\pi(\mathbf{s}_1, \mathbf{a}_1) \right]] \right. \\
&= \mathbb{E}_{\mathbf{s}_1} \left[ r_0 + \gamma(\mathcal{H}(\hat{\pi}(\mathbf{s}_1, \cdot)) + r_1) \right] + \gamma^2 \mathbb{E}_{\mathbf{s}_2} \left[ \mathcal{H}(\pi(\mathbf{s}_2, \cdot)) + \mathbb{E}_{\mathbf{a}_2 \sim \pi} \left[ Q^\pi(\mathbf{s}_2, \mathbf{a}_2) \right] \right] \\
&\leq \mathbb{E}_{\mathbf{s}_1} \left[ r_0 + \gamma(\mathcal{H}(\hat{\pi}(\mathbf{s}_1, \cdot)) + r_1) \right] + \gamma^2 \mathbb{E}_{\mathbf{s}_2} \left[ \mathcal{H}(\hat{\pi}(\mathbf{s}_2, \cdot)) + \mathbb{E}_{\mathbf{a}_2 \sim \hat{\pi}} \left[ Q^\pi(\mathbf{s}_2, \mathbf{a}_2) \right] \right] \\
&= \mathbb{E}_{\mathbf{s}_1, \mathbf{s}_2, \mathbf{a}_2 \sim \hat{\pi}} \left[ [r_0 + \gamma(\mathcal{H}(\hat{\pi}(\mathbf{s}_1, \cdot)) + r_1) + \gamma^2(\mathcal{H}(\hat{\pi}(\mathbf{s}_2, \cdot)) + r_2) \right. \\
&\quad + \gamma^3 \mathbb{E}_{\mathbf{s}_3} \left[ \mathcal{H}(\hat{\pi}(\mathbf{s}_3, \cdot)) + \mathbb{E}_{\mathbf{a}_3 \sim \hat{\pi}} \left[ Q^\pi(\mathbf{s}_3, \mathbf{a}_3) \right] \right] \\
&\quad\quad \vdots \\
&\leq \mathbb{E}_{\tau \sim \hat{\pi}} \left[ r_0 + \sum_{t=1}^\infty \gamma^t (\mathcal{H}(\hat{\pi}(\mathbf{s}_t, \cdot)) + r_t) \right] \\
&= Q^{\hat{\pi}}(\mathbf{s}, \mathbf{a}). \tag{6}
\end{aligned}$$

So when we start with an arbitrary policy $\pi_0$ and define the *policy iteration* as:

$$\pi_i(\mathbf{s}, \mathbf{a}) = \frac{\exp\left(\frac{1}{\alpha} Q^{\pi_i}(\mathbf{s}, \mathbf{a})\right)}{\int_{\mathbf{a}' \in \mathcal{A}} \exp\left(\frac{1}{\alpha} Q^{\pi_i}(\mathbf{s}, \mathbf{a}')\right) d\mathbf{a}'}. \tag{7}$$

Then $Q^{\pi_i}(\mathbf{s}, \mathbf{a})$ optimized monotonically, so $\pi_i$ will converge to $\pi_\infty$, and the optimal policy must satisfy this energy-based Boltzmann distribution.

## 2 TRANSFORMER-BASED SYMBOLIC TREE GENERATOR

We generate the symbolic tree by predicting the predicates of the rules. Given the state-action trajectory and the currently generated partial abstract symbolic tree, our model can calculate the probabilities of the predicate to expand this node, as shown in Eq. (6) in the main paper.

### 2.1 ABSTRACT SYMBOLIC TREE READER

We design an abstract symbolic tree reader to model the structure of the generated partial symbolic tree. While our trees are generated by predicting sequences of predicates, these predicates alone lack a concrete representation of the tree's structure, making them insufficient for predicting the next predicate. Therefore, we apply the abstract symbolic tree reader to incorporate both the predicted predicates and the tree's structural information. It contains a stack of blocks, with the first block containing three distinct sub-layers previously introduced: self-attention, the gating mechanism, and the convolution layer. A residual connection is employed between each pair of consecutive sub-layers, following the approach outlined in He et al. (2016), and is subsequently followed by layer normalization.

**Self-Attention** Within our Transformer block, multi-head attention is utilized to effectively capture long-range dependencies and facilitate the learning of non-linear features. In the case of a sequence of mapping predicates denoted as $X_{(1)}, X_{(2)}, \ldots, X_{(n)}$, their embeddings are obtained through a lookup table. Additionally, positional encoding is employed to encode positional information, which is computed as follows:

$$p_{j,i}[2k] = \sin\left(\frac{i+j}{10000^{2k/j}}\right), \tag{8}$$

$$p_{j,i}[2k+1] = \sin\left(\frac{i+j}{10000^{2k/j}}\right), \tag{9}$$

Here, $p_{i,j}[\cdot]$ refers to a specific dimension within the vector $\mathbf{p}_{i,j}$. In this context, $j$ represents the $j$th block and $k$ represents the embedding size. In the initial reader block, the input consists of the sum of the table-lookup embedding and the position embedding. In subsequent blocks, the input is the vector sum of the lower Transformer block's output and the position embedding specific to that block. The self-attention mechanism employed here follows the same architecture as described in the original Transformer Vaswani et al. (2017). We denote the output of the self-attention as $\mathbf{X}_{(1)}^{\text{self}}, \mathbf{X}_{(2)}^{\text{self}}, \ldots, \mathbf{X}_{(n)}^{\text{self}}$.

**Gating Mechanism** Character embeddings are incorporated after self-attention, and the softmax weight $\mathbf{k}_{(i)}^{(c)}$ for character embeddings is obtained through a transformation from character embedding $\mathbf{X}_{(i)}$. The softmax weight $\mathbf{k}_{(i)}^{(y)}$ for the Transformer's output is derived from a linear transformation of $\mathbf{X}_{(i)}^{\text{self}}$. Additionally, the control vector $\mathbf{q}_{(i)}$ is obtained from $\mathbf{X}_{(i)}^{\text{self}}$ through a linear transformation. The gate can be computed as follows:

$$[\alpha_{(i),t}^{(y)}, \alpha_{(i),t}^{(c)}] = \text{softmax}\{\mathbf{q}_{(i)}^{\text{T}} \mathbf{k}_{(i)}^{(y)}, \mathbf{q}_{(i)}^{\text{T}} \mathbf{k}_{(i)}^{(c)}\}, \tag{10}$$

$\alpha_{(i),t}^{(y)}$ and $\alpha_{(i),t}^{(c)}$ are used to weigh the features of the Transformer's layer $\mathbf{c}_{(i)}^{(y)}$ and the features of character embeddings $\mathbf{c}_i^{(c)}$, which are transformed from $\mathbf{X}_{(i)}^{\text{self}}$ and $\mathbf{X}_{(i)}^{(c)}$, respectively. So the output of

gating can computed as follows:

$$\mathbf{g}_{(i),t} = [\alpha_{(i),t}^{(y)}\mathbf{c}_{(i)}^{(y)}, \ldots, \alpha_{(i),t}^{(c)}\mathbf{c}_{(i)}^{(c)}] \tag{11}$$

$$\mathbf{g}_{[(i),\cdot]} = [\mathbf{X}_{(1)}^{\text{gate}}, \cdots, \mathbf{X}_{(n)}^{\text{gate}}], \tag{12}$$

**Convolution**   Following the gating process, two convolutional layers are utilized to capture local features surrounding each predicate and produce the following output:

$$\mathbf{X}^{(\text{conv},l)} = W^{(\text{conv},l)}[\mathbf{X}_{(i-(w-1)/2)}^{(\text{conv},l-1)}; \cdots; \mathbf{X}_{(i+(w-1)/2)}^{(\text{conv},l-1)}]. \tag{13}$$

Here, $l$ represents the convolutional layer, and $w$ denotes the window size. $\mathbf{X}^{(\text{conv},l)}$ corresponds to the output of the $l$-th convolutional layer. It is important to note that the input to the first layer is the output of the gating process, denoted as $\mathbf{X}_{(i)}^{\text{gate}}$.

To encompass various aspects of information, we represent the tree as a sequence of predicates. We then encode the rules using an attention mechanism and subsequently employ a tree convolution layer to amalgamate the encoded representation of each node with its ancestors. Suppose we have a sequence of predicates $X_{(1)}, X_{(2)}, \ldots, X_{(P)}$, where $P$ denotes the sequence's length. Within the Abstract Symbolic Tree Reader, we generate four types of embeddings:

*Predicate sequence embedding.*   To encode the information of predicates, we use table-lookup embeddings to present these $P$ predicates as real-valued vectors $\mathbf{X}_{(1)}, \mathbf{X}_{(2)}, \ldots, \mathbf{X}_{(P)}$.

*Predicate definition embedding.* The former embedding represents the predicates as an atomic token and loses the information of the predicates' content, so we introduce predicate definition embedding here. For a symbolic predicate $i : \alpha \rightarrow \beta_1, \ldots, \beta_K$, where $\alpha$ is the parent node and $\beta_1, \ldots, \beta_K$ are child nodes (which can be terminal or non-terminal symbols), the index $i$ is the predicate's ID. We encode the predicate content as a vector $\mathbf{X}_c$ using a fully connected layer with inputs being the table-lookup embeddings $\boldsymbol{\alpha}, \boldsymbol{\beta}_1, \cdots, \boldsymbol{\beta}_K$ of the respective symbols, and the sequence is padded to a maximum length. The predicate definition features $\mathbf{X}_{(1)}^{(p)}, \mathbf{X}_{(2)}^{(p)}, \ldots, \mathbf{X}_{(P)}^{(p)}$ are then computed by another fully-connected layer as follows:

$$\mathbf{X}_{(i)}^{(p)} = W^{(p)}[\mathbf{X}_{(i)}; \mathbf{X}_c; \boldsymbol{\alpha}]. \tag{14}$$

Here, $\mathbf{X}_{(i)}$ represents the table-lookup embedding of the predicate $X_{(i)}$ in the symbolic tree, while $\mathbf{X}_c$ represents the content-encoded predicate representation.

*Position embeddings.* Position embeddings are computed as in Eq. (8), representing the position of each predicate within the sequence $X_{(1)}, X_{(2)}, \ldots, X_{(P)}$.

*Depth embeddings.*  As position embeddings may not capture the position of a predicate within the symbolic tree, we introduce depth embedding. Similar to predicate definition embedding, we represent the depth of the predicate based on its parent node without the content embedding.

These embeddings are input into the reader, and after passing through four distinct sub-layers, they are transformed into $\mathbf{X}_{(1)}^{(ast)}, \mathbf{X}_{(2)}^{(ast)}, \ldots, \mathbf{X}_{(P)}^{(ast)}$. In contrast to the first block, we incorporate a cross-attention sub-layer and transform the convolution layer into a tree convolution layer. The cross-attention sub-layer is informed of the input trajectory, facilitated by multi-head attention. The tree-convolution layer is used to amalgamate information about a node and its ancestors. Traditional Transformer architectures struggle to maintain the relationship between two nodes that are far apart in the rule but close in structure. Further details are shown below.

**Cross-Attention**   Incorporating information from the input trajectory is essential. Therefore, we involve the output of the trajectory reader here. This is achieved through a multi-head attention mechanism, following the same approach as the attention mechanism in the Transformer decoder's attention to its encoder.

**Tree Convolution**   Utilizing a traditional convolutional layer to effectively amalgamate information from a node with its ancestors poses challenges. To address this issue, we treat the symbolic tree as a graph and employ an adjacency matrix denoted as $M$ to represent the directed relationships within

the graph. When one predicate $X_{(i)}$ serves as the parent of $X_{(j)}$, it is represented by $M_{(ji)} = 1$. Assuming the outputs of the preceding layer are $\mathbf{X}_{(1)}, \ldots, \mathbf{X}_{(P)}$, we can ascertain the parents of these nodes through matrix multiplication with $M$:

$$[\mathbf{X}_{(1)}^{(\text{parent})}, \cdots, \mathbf{X}_{(P)}^{(\text{parent})}] = [\mathbf{X}_{(1)}, \cdots, \mathbf{X}_{(P)}]M. \tag{15}$$

Here, $\mathbf{X}_{(i)}^{(\text{parent})}$ represents the parent of the $i$th node. It's important to note that the father of the root node is the padded root node. The tree-based convolution window applied to the current sub-tree is given by:

$$\mathbf{X}^{(\text{tconv},l)} = f(W^{(\text{tconv},l)}\left[\mathbf{X}^{(\text{tconv},l-1)}; \mathbf{X}^{(\text{tconv},l-1)}; \ldots; \mathbf{X}^{(\text{tconv},l-1)}M^{w-1}\right]). \tag{16}$$

where $W^{(\text{tconv},l)}$ is the wighrs of the convolutional layer. and $w$ is the window size. $l$ is the layer of these convolutional layers. Similar to the convolution layer in the trajectory reader, the input of the first tree convolution layer is the output of the attention layer.

## 2.2 DECODER OF SYMBOLIC TREE GENERATOR

Our final component is a decoder that integrates information from generated logic rules with the state-action trajectory description and predicts the next predicate. It consists of a stack of blocks, each containing several sub-layers. Each sub-layer is surrounded by a residual connection followed by layer normalization. The decoder treats the non-terminal node to be expanded as a query, represented as a path from the root to the node to be expanded. These nodes in the path are represented as real-valued vectors, then a fully connected layer is applied to these vectors and outputs a path of the symbolic tree. Then two attention layers were applied to integrate the outputs of the first block $\mathbf{X}_{(1)}^{(sat)}, \mathbf{X}_{(2)}^{(sat)}, \ldots, \mathbf{X}_{(n)}^{(sat)}$ and the tree convolutional block $\mathbf{X}_{(1)}^{(ast)}, \mathbf{X}_{(2)}^{(ast)}, \ldots, \mathbf{X}_{(P)}^{(ast)}$. Finally, two fully connected layers were used to extract features for prediction.

## 3 Q-FUNCTION WITH RESPECT TO THE ANCHOR ACTION

To begin, let's isolate the case where $t = 0$ from the summation in the value function and derive the following expression:

$$V(\mathbf{s}) = \mathbb{E}\left[r(\mathbf{s}, \mathbf{A}_0) + \alpha\mathcal{H}\left(\pi^*\left(\mathbf{s}, \cdot\right)\right)\right] + \sum_{t=1}^{\infty} \gamma^t \mathbb{E}\left[r\left(\mathbf{S}_t, \mathbf{A}_t\right) + \alpha\mathcal{H}\left(\pi^*\left(\mathbf{S}_t, \cdot\right)\right)\right]. \tag{17}$$

By the definition of Q-function in the Eq. (2), we can get:

$$V(\mathbf{s}) = \mathbb{E}\left[Q\left(\mathbf{s}, \mathbf{A}\right)\right] + \alpha\mathcal{H}\left(\pi^*\left(\mathbf{s}, \cdot\right)\right). \tag{18}$$

where the expectation is over the action following the optimal policy Eq. (2) in the main paper. Next, by the definition of expectation and information entropy, we can derive

$$
\begin{aligned}
V(\mathbf{s}) &= \int_{\mathbf{a}\in\mathcal{A}} Q\left(\mathbf{s}, \mathbf{a}\right)\pi^*\left(\mathbf{s}, \mathbf{a}\right)d\mathbf{a} - \alpha\int_{\mathbf{a}\in\mathcal{A}} \log\left(\pi^*\left(\mathbf{s}, \mathbf{a}\right)\right)\pi^*\left(\mathbf{s}, \mathbf{a}\right)d\mathbf{a} \\
&= \int_{\mathbf{a}\in\mathcal{A}} Q\left(\mathbf{s}, \mathbf{a}\right)\pi^*\left(\mathbf{s}, \mathbf{a}\right)d\mathbf{a} - \alpha\int_{\mathbf{a}\in\mathcal{A}} \frac{Q\left(\mathbf{s}, \mathbf{a}\right)}{\alpha}\pi^*\left(\mathbf{s}, \mathbf{a}\right)d\mathbf{a} \\
&\quad + \alpha\int_{\mathbf{a}\in\mathcal{A}} \log\left[\int_{\mathbf{a}'\in\mathcal{A}} \exp\left(\frac{Q\left(\mathbf{s}, \mathbf{a}'\right)}{\alpha}\right)d\mathbf{a}'\right]\pi^*\left(\mathbf{s}, \mathbf{a}\right)d\mathbf{a} \\
&= \alpha\log\left[\int_{\mathbf{a}'\in\mathcal{A}} \exp\left(\frac{Q\left(\mathbf{s}, \mathbf{a}'\right)}{\alpha}\right)d\mathbf{a}'\right] \\
&= \alpha\log\left[\int_{\mathbf{a}\in\mathcal{A}} \exp\left(\frac{Q\left(\mathbf{s}, \mathbf{a}\right)}{\alpha}\right)d\mathbf{a}\right].
\end{aligned} \tag{19}
$$

Then, we consider a anchor action $\mathbf{a}^A$, and extract $\alpha \log \left[ \exp \left( \frac{Q(\mathbf{s}, \mathbf{a}^A)}{\alpha} \right) \right]$ from Eq. (19):

$$
\begin{aligned}
V(\mathbf{s}) &= \alpha \log \left[ \frac{\int_{\mathbf{a} \in \mathcal{A}} \exp \left( \frac{Q(\mathbf{s}, \mathbf{a})}{\alpha} \right) d\mathbf{a}}{\exp \left( \frac{Q(\mathbf{s}, \mathbf{a}^A)}{\alpha} \right)} \right] + \alpha \log \left[ \exp \left( \frac{Q \left( \mathbf{s}, \mathbf{a}^A \right)}{\alpha} \right) \right] \\
&= \alpha \log \left( \frac{1}{\pi^* \left( \mathbf{s}, \mathbf{a}^A \right)} \right) + Q \left( \mathbf{s}, \mathbf{a}^A \right) \\
&= -\alpha \log \left( \pi^* \left( \mathbf{s}, \mathbf{a}^A \right) \right) + Q \left( \mathbf{s}, \mathbf{a}^A \right).
\end{aligned}
\tag{20}
$$

Then according to the Theorem 2. in Haarnoja et al. (2017), we have

$$
Q(\mathbf{s}, \mathbf{a}) = r(\mathbf{s}, \mathbf{a}) + \gamma \mathbb{E} \left[ V(\mathbf{s}') \right].
\tag{21}
$$

Finally, by taking Eq. (20) into Eq. (21), we get the connection:

$$
Q(\mathbf{s}, \mathbf{a}) = r(\mathbf{s}, \mathbf{a}) + \gamma \mathbb{E}_{\mathbf{s}'} \left[ -\alpha \log(\pi^*(\mathbf{s}', \mathbf{a}^A)) + Q(\mathbf{s}', \mathbf{a}^A) \mid \mathbf{s}, \mathbf{a} \right].
\tag{22}
$$

## 4 DATASET DESCRIPTION

**BlocksWorld.** In this environment, the agent will learn how to stack the blocks into certain styles, that are widely used as a benchmark problem in the relational reinforcement learning research. The blocks world environment contains two worlds: the initial world and the target world, each containing the ground and $m$ blocks. The task is to take actions in the operating world and make its configuration the same as the target world. The agent receives positive rewards only when it accomplishes the task and the sparse reward setting brings significant hardness.

**MIMIC Dataset.** We consider the Medical Information Mart for Intensive Care (MIMIC-III) database and MIMIC-IV (Johnson et al., 2023) database to predict prescription based on 8 observations – temperature, white blood cell count, heart rate, hematocrit, hemoglobin, blood pressure, creatinine, and potassium. MIMIC-III contains 7,493 patients with multiple visits from 2001 to 2012, while there are 85,155 patients in MIMIC-IV with multiple visits from 2008 to 2019. Since there is an overlapped time range between MIMIC-III and MIMIC-IV, we randomly sampled 10,000 patients from MIMIC-IV from 2013 to 2019. By the nature of real-world clinical practice, observation history must be considered by the acting policies – making our decision-making environments partially observable.

**Sorting.** This task trends to iterative swap elements to sort the array in ascending order. Given a length-$m$ array $a$ of integers, We treat each slot in the array as an object and input their index relations and numeral relations to each model.

**Finding Path.** Given an undirected graph represented by its adjacency matrix as relations, the algorithm needs to find a path from a start node to the target node. We formulate the shortest path task as a decision-making task. The agent iteratively chooses the next node along the path. In the next step, the starting node will become the next node.