# OpenReview forum: "Discovering Logic-Informed Intrinsic Rewards to Explain Human Policies"
_ICLR.cc/2024/Conference — Submitted to ICLR 2024_

### Official Review · Reviewer_1taW · 2023-10-29

**Soundness:** 2 fair
**Presentation:** 1 poor
**Contribution:** 3 good
**Rating:** 6
**Confidence:** 3

**Summary:**

The paper proposed an inverse reinforcement learning method to discover a logic-informed reward function. Assuming demonstrations were generated by experts following an optimal energy-based policy, it alternates between learning a neural logic tree and learning policy until convergence with a GAN-framework from a previous work. The energy function is parameterized using logic-informed features given a set of generated logic rules. Specifically, a transformer-based reader encodes the observed state-action demonstrations to a set of Boolean logic variables (predicates), then a decoder predicts the next predicate based on the previously generated partial symbolic tree. Experiments on toy games and two highly correlated real healthcare datasets show overall improvement from several benchmarks under two metrics each.

**Strengths:**

1. proposed a novel approach to IRL.
2. the way the logic rules inform the energy function and consequently the policy makes sense to me.

**Weaknesses:**

1. lack of clarity and justification. E.g., why is the traversal pre-order, how is the set of predefined labeling functions predefined, what is grounded predicate sequence, why grounded predicate is divided into characters by the 'first' block of the abstract symbolic tree reader. If the tree is based on transformer, perhaps the authors could focus more on how the tree is built upon transformer and the difference between the two. See more major ones in my questions.
2. since the contribution is on logic-informed IRL, would be good to show and analyze the logic rules discovered along a trajectory as opposed to a single snapshot.

**Questions:**

1. what is the numerical form of the tree? Eq. 6 only specifies the likelihood of the tree in terms of pre-order traversal sequence, but not the structure thereof (i.e. parent/children nodes).
2. are the cardinalities of the index sets $I_k^1$ and $I_k^0$ same across $k$?
3. how is the node chosen for expansion and how is it expanded? By what criterion you'd know the tree cannot be further expanded?
4. does the order of the predicates generated matter? at the same tree horizon?
5. how is the goal $X^0$ decided, is it always success v. fail? Is it constant throughout a trajectory or changing over timesteps? If it's the former how can you make sure it is the right goal for all timesteps (one step of mishap would not necessarily reasult in an overall failure and vice versa), if the latter then how is the step-wise goal specified without too much human knowledge?
6. in estimating the overall energy function, since the tree generator is amortized, why you can use the top-K logic trees with generated probabilities as the probabilities may change later? And can you not use unweighted trees (e.g. taking average of all trees) to approximate the expectation?

---

> ### Author Response · Authors · 2023-11-21
> **Rebuttal by Authors**
>
> **Q1. Why is the traversal pre-order?**
>
> A1:The choice of pre-order traversal for processing the logic tree is guided by its inherent property to prioritize root nodes before leaves. This sequence aligns with how our model hierarchically processes predicates, ensuring that higher-level decisions inform the evaluation of subsequent conditions. Pre-order traversal is well-suited for generating a sequence of predicates that reflects the structured reasoning our model embodies, mirroring human decision-making patterns where fundamental judgments precede more detailed considerations. Moreover, this approach is consistent with methodologies used in previous research [1], particularly in the context of natural language processing.
>
> [1]Sun, Zeyu, et al. "Treegen: A tree-based transformer architecture for code generation." Proceedings of the AAAI Conference on Artificial Intelligence. 2020.
>
> **Q2. How is the set of predefined labeling functions predefined?**
>
> A2: The labeling functions are predefined based on a combination of domain expertise and pretrained model. Domain experts identify relevant features that could potentially serve as informative predicates. These are then validated against the data to ensure they are significant for the model's predictive performance. We implement an iterative process where the initial set of labeling functions is refined through model feedback, emphasizing the most discriminative features for the logic tree construction. This ensures that our predefined labeling functions are both theoretically sound and empirically robust, leading to the creation of a logic tree that is well-founded and effective for the task at hand.
>
> **Q3. What is a grounded predicate sequence?**
>
> A3: A grounded predicate sequence in logic trees is a sequence where each predicate has specific, often binary, values assigned to its variables. This transforms abstract logic rules into concrete, evaluable conditions. In the context of a logic tree, this sequence forms a distinct path marked by predicates grounded with actual data. It's crucial for model interpretability, as it offers a clear and logical flow of deductions leading to a specific outcome.
>
> **Q4.Why grounded predicate is divided into characters by the 'first' block of the abstract symbolic tree reader.**
>
> A4: The division of grounded predicates into characters within the 'first' block of our abstract symbolic tree reader is a strategy to encapsulate raw trajectory data, specifically state and action information, into the logic tree structure. This process facilitates the embedding of concrete trajectory instances, preserving the integrity of state/action sequences within the decision-making framework.
>
> **Q5: If the tree is based on transformer, perhaps the authors could focus more on how the tree is built upon transformer and the difference between the two.**
>
> A5: Our logic tree, while drawing inspiration from transformer architectures, is specially crafted to enhance interpretability in decision-making. Transformers adeptly map global dependencies, yet our logic tree is devised to present decisions in a hierarchical and intelligible fashion. This tailored design facilitates transparent and traceable reasoning, essential in critical domains. Additionally, our supplementary material details a transformer architecture incorporating a tree convolution layer. This layer is pivotal in grasping the contextual relationships of nodes within the logic tree, marking a key innovation from traditional transformer models.
>
> **Q6: Since the contribution is on logic-informed IRL, would be good to show and analyze the logic rules discovered along a trajectory as opposed to a single snapshot.**
>
> A6: Thank you very much for your valuable recommendation. You raise an important point regarding the potential insights that could be gained from analyzing the evolution of logic rules along a trajectory in our logic-informed IRL study. Currently, our methodology centers around a logic tree that primarily captures static conjunctions ('and' relations) among various logical components. Our predicates are not time-based, which means that our current framework might not be ideally suited to capture the dynamic evolution of these rules over time. However, your suggestion has certainly inspired us to consider the possibilities of integrating spatio-temporal logic rules in our future research.

---

> > ### Author Response · Authors · 2023-11-21
> > **Rebuttal by Authors**
> >
> > **Q7: What is the numerical form of the tree? Eq. 6 only specifies the likelihood of the tree in terms of pre-order traversal sequence, but not the structure thereof (i.e. parent/children nodes).**
> >
> > A7: The numerical form of the logic tree is represented in two ways:
> > - Conjunction Clauses (Eq. 5): The logic tree comprises a set of logic rules, each written as a conjunction clause. These rules include positive and negative predicates, which are the conditions required to reach the goal.
> > - Energy Function Representation (Eq. 7): The logic tree is also conceptualized as an energy function. In this representation, trained parameters   can be interpreted as the weights of each rule in the logic tree. This representation captures the tree's numerical form, integrating both the structural logic and the significance of each rule within the framework.
> >
> > **Q8: are the cardinalities of the index sets $I_k^1$ and $I_k^0$ same across k?**
> >
> > A8: The cardinalities of  $I_k^1$ and $I_k^0$ indeed vary across different k. Specifically, $I_k^1$ corresponds to the indices of positive predicates within a rule, whereas $I_k^0$ relates to indices of negative predicates. This variability is intrinsic to the design of our rules and reflects the distinct conditions each rule captures within our logic-informed framework.
> >
> > **Q9: how is the node chosen for expansion and how is it expanded? By what criterion you'd know the tree cannot be further expanded?**
> >
> > A9: In our model, node selection for expansion is informed by the current state of the partial logic tree and input data. The expansion involves a classification process for predicates, where a decoder assesses the likelihood of each option. We also employ a pointer network to directly identify a predicate from the input, designating it as a logic tree's terminal node. Importantly, a gating mechanism, derived from the decoder's last feature, guides the decision to either expand or terminate, ensuring that expansion aligns with the tree’s current state and the available data.
> >
> > **Q10: does the order of the predicates generated matter? at the same tree horizon?**
> >
> > A10: Thank you for your question. In our logic tree, the order of predicates at the same level does not affect the outcome since each is connected by a logical "AND" conjunction. The tree's structure dictates that the satisfaction of the root node condition relies on the collective satisfaction of all its associated leaf nodes, regardless of their order. This design ensures that the interpretability of the tree's decision-making process remains clear and logically sound.
> >
> > **Q11: How is the goal $X^0$ decided, is it always success v. fail? Is it constant throughout a trajectory or changing over timesteps? If it's the former how can you make sure it is the right goal for all timesteps (one step of mishap would not necessarily reasult in an overall failure and vice versa), if the latter then how is the step-wise goal specified without too much human knowledge?**
> >
> > A11: The goal $X^0$ in our model is conceptualized based on the desired outcome of the task, which may indeed be as binary as success versus failure in certain scenarios. It is generally fixed throughout a trajectory to provide a consistent objective. However, we understand that a single mishap does not necessarily dictate the final outcome, and our model accounts for this by considering the cumulative progress towards the goal, rather than stepwise success or failure.
> >
> > **Q12: in estimating the overall energy function, since the tree generator is amortized, why can you use the top-K logic trees with generated probabilities as the probabilities may change later? And can you not use unweighted trees (e.g. taking average of all trees) to approximate the expectation?**
> >
> > A12: In our model, the top-K logic trees are selected based on their current probabilities. This selection reflects a snapshot of the most probable trees at a given iteration, which contributes to a more tractable optimization process. We acknowledge that probabilities may evolve; however, this approach allows us to approximate the expectation without the computational expense of considering all possible trees. As for using unweighted trees, it could lead to a less accurate estimate since it would not account for the varying significance of different trees. We prioritize weighted contributions to capture the nuances, which is more effective.

---

> > > ### Comment · Reviewer_1taW · 2023-11-21
> > >
> > > Thank you authors for your response. I acknowledge that most of my concerns have been addressed.
> > >
> > >
> > > Trivial further questions:
> > >
> > > in A10, if the order of predicates at the same level is irrelevant, how would you ensure the pre-order traversal is insensitive to it?
> > >
> > > in A11, how would you compute the cumulative progress towards the goal?
> > >
> > >
> > > I am raising my score. However, the method depends largely on human picked features which could be multitudinous, leading to a trade-off between inaccuracy and human involvement.

---

> ### Author Response · Authors · 2023-11-22
> **Rebuttal by Authors**
>
> Thank you for your insightful inquiries and interest in our research. We are pleased to provide further clarification on your questions as follows:
>
> **Q13: If the order of predicates at the same level is irrelevant, how would you ensure the pre-order traversal is insensitive to it?**
>
> A13: To ensure that pre-order traversal is insensitive to the order of predicates at the same level in a tree structure, the use of position and depth embeddings is key. We have introduced them in the section 2 of our supplementary file. Here's how it works:
>
> **1.Position Embeddings**: These assign a unique identifier to each node based on its position in the pre-order traversal. This helps in capturing the sequence information of the nodes.
>
> **2.Depth Embeddings**: These embeddings encode the depth of each node in the tree. Nodes at the same level will have identical depth embeddings, making the representation insensitive to their order.
>
> By combining position and depth embeddings, the representation of each node incorporates both its sequence in the traversal and its hierarchical level in the tree. This approach ensures that the tree representation remains consistent regardless of the order of nodes at the same level, focusing instead on their structural positions.
>
> **Q14: how would you compute the cumulative progress towards the goal?**
>
> A14: In our framework, the primary stages are logic tree learning and policy learning, which mirror backward and forward reasoning, respectively. The neural logic tree generator sequentially composes logic variables, starting from the goal, to establish strategic rules. In the policy learning stage, the agent is optimized to follow the most favorable path for forward chaining, as dictated by these rules.** These stages are synergistically linked by a neural symbolic energy function, denoted as Eq. (7). This function serves as a critical intermediary, assessing the extent to which rule conditions are met and guiding the GAN in refining the policy learning process**. Specifically, **each summand in Eq. (7) acts as a soft approximation of a clause in the logic tree**, facilitating a nuanced and flexible approach to rule application. Through an iterative process of alternating between logic tree generation and policy optimization, our framework progressively converges on the most effective set of logical rules and corresponding policies.

---

### Official Review · Reviewer_5AXy · 2023-11-01

**Soundness:** 3 good
**Presentation:** 3 good
**Contribution:** 3 good
**Rating:** 5
**Confidence:** 4

**Summary:**

In this paper, a novel logic-informed Inverse Reinforcement Learning (IRL) framework is introduced. The approach embodies inverse optimal control through policy optimization, where logic rules are learned from expert trajectories and serve as the energy function. The policy is optimized to estimate the energy model's partition function. Essentially, the policy is trained to generate state-action trajectories that minimize the energy function encoded from the currently learned logic rules, ensuring better adherence to the logic rules. The paper employs a GAN-style training scheme to update these logic rules by discerning trajectories generated by the policy from expert trajectories. The framework utilizes a neural logic tree generator to sequentially derive logic rules from goal variables, mimicking backward reasoning, and employs policy learning to determine the most effective path to achieve the end goal based on the current logic rules, akin to forward reasoning. This alternating process of backward and forward reasoning continues until convergence is attained, enabling the method to potentially learn the optimal probabilistic distribution of logic trees and the policy.

**Strengths:**

* Interpretability: The paper introduces a novel Inverse Reinforcement Learning (IRL) framework that learns both logical reasoning processes employed by experts and policies from observational data. This dual-learning approach improves policy interpretability, distinguishing it from traditional black-box solutions. Logic rules learned from this framework can be used to explain the observational state-action trajectories from expert demonstrations.

* Reward Recovery: The paper introduces a reward learning framework that appears to be both manageable and effective. This framework facilitates the automatic exploration of intrinsic logical knowledge, as manifested in the symbolic logic trees implicitly employed by experts for guiding reward design.

* The experiment results regarding policy and logic rules learning seem convincing.

**Weaknesses:**

I have a positive view of this work. However, the reason for not assigning a positive score to the paper lies in the absence of a clear evaluation of the reward discovery aspect (as mentioned in Section 4.2, a claimed contribution of this paper). While Section 4.2 provides informative content and closely follows the prior work of deep PQR (Geng et al., 2020b), the central focus appears to be on the relationship between the discovered rewards and the learned logic rules. Unfortunately, the paper lacks concrete examples or evaluations to support this argument. Particularly, there is a lack of evaluation regarding whether it is possible to predict the decision-making of the experts using the estimated reward functions on benchmark tasks.

Similarly, there has been no evaluation conducted on the quality of the learned logic rules.

**Questions:**

For practical application of this approach, users are required to define predicate sets beforehand to facilitate the learning of logic rule-informed energy functions. I am curious whether there has been an ablation study conducted to assess the algorithm's performance concerning the quality and suitability of the provided predicates.

Is there any evaluation regarding predicting the decision-making of experts using the estimated reward functions on your benchmarks?

Could you assess the precision and recall of your learned logic rules in comparison to expert decisions on the provided benchmarks?

Why do Sec 5.1 and Sec 5.2 use different baselines?

Was the evaluation conducted in a fair manner for all the baselines? For instance, NLRL does not utilize expert trajectories and learns directly from an MDP. In contrast, your approach benefits from access to expert trajectories.

---

> ### Author Response · Authors · 2023-11-21
> **Rebuttal by Authors**
>
> **Q1: I have a positive view of this work. However, the reason for not assigning a positive score to the paper lies in the absence of a clear evaluation of the reward discovery aspect...**
>
> A1: Thanks for your suggestions. To evaluate the estimated reward functions, we will compare the decisions made by experts with the decisions predicted by the estimated reward functions. Although our benchmarks do not contain the decisions made by experts, we follow [1] to evaluate our method against GAIL and AIRL on the Swimmer dataset. We predict the decision-making using the estimated reward function. Numerical results are presented in the following table. Our method can also obtain comparable scores in the Swimmer dataset.
>
> Table1: Results on swimmer tasks. Negative log-likelihood (lower is better) are reported across 5 runs.
>
> |  Methods   | Negative log-likelihood  |
> | :---  | ---:  |
> | AIRL | 0.39 |
> | GAIL | 0.34 |
> | Ours | 0.26 |
>
> [1] Fu J, Luo K, Levine S. Learning robust rewards with adversarial inverse reinforcement learning[J], 2017.
>
>
> **Q2: Similarly, there has been no evaluation conducted on the quality of the learned logic rules.**
>
> A2: Thanks for your advice. In the real-world dataset, there is no ground-truth of learned logic rules, so we follow [3] to verify our model’s rule discovery ability on synthetic datasets with a known set of ground-truth rules and weights. Note that it was originally utilized for the temporal point process, so we modify it by adding spatial variables (such as “left, right, front, and behind”) to fit in our settings. The weight learning results on 4 synthetic datasets are shown in the following table.
>
> Table2: Rule discovery and weight learning results (GT weights/learned weights) on 4 synthetic datasets.
>
> |  Weights   | Dataset-1  | Dataset-2  | Dataset-3  | Dataset-4  |
> | :---  | ---:  | ---:  | ---:  | ---:  |
> | w0| 1.00/0.98 | 0.50/0.45 | 1.50/1.47 | 2.00/1.82 |
> | w1| 1.00/0.91 | 0.50/0.40 | 1.50/1.44 | 1.00/0.97 |
> | w2| 1.00/0.81 | 0.50/0.34 | 1.50/1.39 | 1.00/0.92 |
>
> [3] Li S, Feng M, Wang L, et al. Explaining point processes by learning interpretable temporal logic rules[C]/ICLR. 2021.
>
> **Q3:I am curious whether there has been an ablation study conducted to assess the algorithm's performance concerning the quality and suitability of the provided predicates.**
>
> A3: Thanks for your suggestions. Our generator takes the observed state-action demonstrations as input, and initially encodes them into the symbolic predicate space. So the input predicate has a great influence on the final results. Moreover, we removed different portions (denoted as p) of the input predicate, and the results are shown in the following table. we can see that when we remove some input predicates, the performance of R@10 and R@20 drops significantly.
>
> Table 3: Ablation study of removing different portions (denoted as p) of the input predicate. Note that p=10% means that we remove 10% predicates from the framework. We show diagnosis prediction results on MIMIC-III using w-F1 (%) and R@k (%).
> |  Metric | p=0%  | p=10%  | p=20%  | p=30%  | p=40% |
> | :---  | ---:  | ---:  | ---:  | ---:  | --:  |
> | R@10 | 29.01(0.11) | 25.32(0.16) | 23.21(0.18) | 19.93(0.08) | 18.24(0.09) |
> | R@20 | 38.10(0.08) |33.45(0.14)  | 32.81(0.07) | 31.64(0.11) | 30.83(0.06) |
>
> **Q4: Is there any evaluation regarding predicting the decision-making of experts using the estimated reward functions on your benchmarks?**
>
> A4: Be the same as Q1.
>
> **Q5: Could you assess the precision and recall of your learned logic rules in comparison to expert decisions on the provided benchmarks?**
>
> A5: Be the same as Q2.
>
> **Q6: Why do Sec 5.1 and Sec 5.2 use different baselines?**
>
> A6: The difference of baselines results from the different datasets. We compare our method with several baselines (e.g., NLM and MemNN) in the Blockworld dataset because their authors have evaluated the performance in the Blockworld dataset in published papers. Moreover, HiTANet and Chet are also representative baselines in the MIMIC dataset, so we compared our method with it to demonstrate our performance in healthcare event prediction and meanwhile provide strategic explanations.
>
> **Q7: Was the evaluation conducted in a fair manner for all the baselines? **
>
> A7: Our evaluation was designed to demonstrate the enhanced learning potential when expert trajectories are incorporated, as opposed to a direct comparison of learning methods. While NLRL learns solely from an MDP, our approach leverages expert trajectories for a more guided learning process. This difference in methodology is crucial for the context of our study, as it highlights the advantages of integrating expert knowledge. The evaluation is thus fair within the scope of our research objectives, which is to showcase the benefits of using expert data in learning algorithms.

---

> ### Comment · Reviewer_5AXy · 2023-11-21
> **Thanks for your response**
>
> I thank the authors' for the detailed response.
>
> **To evaluate the estimated reward functions, we will compare the decisions made by experts with the decisions predicted by the estimated reward functions.**
>
> The presented results in the rebuttal is on the swimmer task, which is not listed in the original paper. I would appreciate it if you could consider conducting experiments with the algorithmic and healthcare benchmarks from the paper (instead of using ad hoc new benchmarks). Doing so would make it possible to integrate the results into the paper.
>
> **Evaluation conducted on the quality of the learned logic rules.**
>
> Similarly, the experiments on the quality of the learned logic rules are not satisfactory. The results were obtained on a new set of benchmarks rather than the benchmarks specified in the paper. Is it possible to evaluate how often the logic rules can deduce the same decision actions as those conducted in the expert demonstrations?
>
> Given the aforementioned concerns, I maintain my opinion that the paper falls below the threshold for acceptance.

---

> > ### Author Response · Authors · 2023-11-22
> > **Rebuttal by Authors**
> >
> > Thank you for your insightful recommendation. We appreciate your perspective and understand the importance of consistency in the benchmarks used for our research.
> >
> > Regarding the application of our methodology to the MIMIC healthcare dataset, we face a significant limitation: the absence of ground truth for the learned logic rules. This constraint precludes us from conducting analogous experiments in this domain as we did with the Swimmer synthetic dataset.
> >
> > To address this challenge and to ensure the validity of our results, we propose an alternative approach. We plan to leverage the expertise of medical professionals or researchers well-versed in this field to interpret and validate our estimated reward function and the logic rules we have learned. Their insights will be instrumental in assessing the rationality and applicability of our findings in a real-world healthcare context.
> >
> > Doctor Verification and Medical Reference: our methodology included the invaluable input of medical experts who reviewed our discovered rule results Their feedback consistently affirmed the sensibility of our findings. Furthermore, we reinforced our discovered rules with evidence from medical references, as exemplified below:
> >
> > SpO2↑ $\leftarrow$ Temperature↓ $\wedge$ SpO2 Low:  After the body temperature drops to the normal range, if the Saturation of Peripheral Oxygen keeps low at this time, the patient takes the treatment to keep SpO2 in the normal range [1].
> >
> > Systolic↓ $\leftarrow$ WBCC↓ $\wedge$ Systolic High: In early stage, the systolic blood pressure serves as an indicator. When the  white blood cell count decreases but the systolic blood pressure keeps high, the patient takes the treatment to keep the systolic blood pressure down, which will assist in the patient’s subsequent recovery [2].
> >
> > We will include these additional steps and their outcomes in our manuscript, providing a comprehensive view of how expert interpretation can complement our methodology when direct experimental validation is not feasible.
> >
> > [1] Mizock B A. Alterations in carbohydrate metabolism during stress: a review of the literature[J]. The American journal of medicine, 1995, 98(1): 75-84.
> >
> > [2] Komorowski M, Celi L A, Badawi O, et al. The artificial intelligence clinician learns optimal treatment strategies for sepsis in intensive care[J]. Nature medicine, 2018, 24(11): 1716-1720.

---

> > > ### Comment · Reviewer_5AXy · 2023-11-22
> > >
> > > Thank you for your reply. I'm curious if it would be feasible to predict the decision-making of experts using the estimated reward function with the benchmarks provided in the paper. The absence of an evaluation like this to assess the quality of the reward functions is currently the main limiting factor for me to support accepting this manuscript.

---

> > > > ### Author Response · Authors · 2023-11-23
> > > > **Rebuttal by Authors**
> > > >
> > > > Thank you for your continued engagement and valuable feedback. We acknowledge your concern regarding the absence of a direct evaluation of the estimated reward functions against expert decision-making in the specific benchmarks mentioned in our paper.
> > > >
> > > > To address this, we have designed an experiment utilizing the blocksworld dataset, which is included in our main paper. In evaluating the estimated reward functions, we adopted a methodology akin to DeepPQR, focusing on calculating the Mean Squared Error (MSE) for reward recovery. A key aspect of our method involves the assumption of an 'anchor action', which is presumed to have a reward of zero. To ensure fairness in our comparisons, we have grounded the reward estimators of other methods at this anchor action, setting their reward to zero as well.
> > > >
> > > > The results from this experiment demonstrate that our method provides accurate estimates of the reward function, compared with NLM. This finding is significant as it underpins the effectiveness of our approach in closely aligning with expert decisions, even within the constraints of our dataset.
> > > >
> > > >
> > > > Table: MSE (lower is better) for reward recovery with a different number of blocks, where m is the number of blocks in the blocks world environment.
> > > >
> > > > | Methods    | m=10     | m=20     | m=30 | m=40 | m=50|
> > > > | -------- | -------- | -------- | -------- | -------- | -------- |
> > > > | NLM | 0.225 | 0.219 | 0.138 | 0.439 | 0.210|
> > > > | Ours | 0.093 | 0.139 | 0.124 | 0.351 | 0.155|

---

### Official Review · Reviewer_Neph · 2023-11-01

**Soundness:** 2 fair
**Presentation:** 2 fair
**Contribution:** 3 good
**Rating:** 6
**Confidence:** 3

**Summary:**

This paper introduces a new technique for Inverse Reinforcement Learning grounded in constructing Logic Trees that can help with interpreting a given expert policy dataset while also yielding the policies deployed by the expert. It does this by means of a neural rule generator that creates a tree in a top down fashion. These trees are modeled to fit the trajectory data by means of a logic-informed energy function which is further combined with a GAN-based framework to determine the parameters matching the data. This is then used to get a policy matching the logic tree following which the reward distribution can be estimated as well. Experimental results show the resulting policies are interpretable while also being more efficient in several RL-based settings. Real-world results on the MIMIC-III and MIMIC-IV datasets also demonstrate sufficient performance while maintaining interpretability.

**Strengths:**

- Logic based tree formulation is simple yet informative on the exact decision making process by the expert policies.
- Strong results in the RL-based experiments (Table 1) showing promise of performance while maintaining interpretability.

**Weaknesses:**

- Results in Table 2 on Diagnosis Prediction may be too close to edge out competing methods (Chet [1]) albeit being more interpretable by means of the Logic Tree.
- It is not entirely clear how to determine the  predicates for any given problem. The quality of these predicates will largely determine the quality of the logic tree and output policy.

References:

[1] Context-aware Health Event Prediction via Transition Functions on Dynamic
Disease Graphs, Lu et al., 2022

**Questions:**

1. How does the algorithm handle redundant sets of nodes in a given Logic tree? Is there a pruning procedure?
2. How are the predicate variables determined? Are the functions of the observation space manually provided by the user for a given dataset? E.g. Above(x,y) in the Blockworld experiments
3. Could there be any additional experiments showing the effect of input predicate set choice on the final result?
4. How are the competing algorithms being shown fairly since they are not provided these informative predicates? Could they be included as part of the observation space and run again?

Minor Typos:

Introduction (paragraph 4) : “Our ILR involves”

Fig. 3 “Sucess”

Fig. 4 “Temperture”

---

> ### Author Response · Authors · 2023-11-21
> **Rebuttal by Authors**
>
> **Q1: Results in Table 2 on Diagnosis Prediction may be too close to edge out competing methods (Chet [1]) albeit being more interpretable by means of the Logic Tree.**
>
> A1:Thank you for your insightful comment regarding the results presented in Table 2. While the performance enhancements over Chet [1] may seem modest, the significant increase in interpretability achieved through our Logic Tree method represents a crucial advancement. In domains where the stakes of decisions are high, the ability to comprehend and trust the decision-making process is invaluable. Our approach underscores the importance of explainable AI, balancing a slight uptick in performance with a marked improvement in transparency. This balance is further supported by [2], which articulates the often necessary trade-off between precision and interpretability in model design.
>
> [1] Context-aware Health Event Prediction via Transition Functions on Dynamic Disease Graphs, Lu et al., 2022
>
> [2] Rudin C. Stop Explaining Black Box Machine Learning Models for High Stakes Decisions and Use Interpretable Models Instead. Nat Mach Intell. 2019.
>
> **Q2: It is not entirely clear how to determine the predicates for any given problem. The quality of these predicates will largely determine the quality of the logic tree and output policy.**
>
> A2: Thank you for your insightful comment regarding predicate determination. In our approach, we rely on domain expertise, which is a widely adopted practice within this field, to guide the selection of predicates. While there is existing research on automated predicate invention, such as Meta-Interpretive Learning (MIL) [3], these methods often face limitations in complex domains like healthcare, where the intricacies of the data require expert knowledge. Predicates crafted with the input of medical professionals ensure reliability and relevance, leading to more meaningful and actionable logic rules.
>
> [3] Muggleton S H. Meta-interpretive learning of higher-order dyadic datalog: Predicate invention revisited[J]. Machine Learning, 2015.
>
> **Q3: How does the algorithm handle redundant sets of nodes in a given Logic tree? Is there a pruning procedure?**
>
> A3: Our algorithm proactively minimizes redundancy in logic trees through the use of expertly crafted predicates, ensuring tree compactness from the outset. Additionally, our tree generator operates within an amortized framework, which facilitates the concurrent optimization of the tree structure and the parameters of an energy-based Generative Adversarial Network (GAN) during policy function estimation. This integrated approach effectively prunes unnecessary nodes, maintaining the efficiency and relevance of the logic tree representation.
>
> **Q4: How are the predicate variables determined?**
>
> A4: Be the same as Q2.
>
> **Q5: Could there be any additional experiments showing the effect of input predicate set choice on the final result?**
>
> A5: Thanks for your suggestions. Our generator takes the observed state-action demonstrations as input, and initially encodes them into the symbolic predicate space. So the input predicate has a great influence on the final results. Moreover, we removed different portions (denoted as p) of the input predicate, and the results are shown in the following table. we can see that when we remove some input predicates, the performance of R@10 and R@20 drops significantly.
>
> Table 1: Ablation study of removing different portions (denoted as p) of the input predicate. Note that p=10% means that we remove 10% predicates from the framework. We show diagnosis prediction results on MIMIC-III using w-F1 (%) and R@k (%).
> |  Metric   | p=0%  | p=10% | p=20% | p=30% | p=40%|
> | :---  | ---:  | :--: |  :--: |  :--: |  :--: |
> | R@10  | 29.01(0.11) |25.32(0.16)|23.21(0.18)|19.93(0.08)|18.24(0.09)|
> | R@20  | 38.10(0.08)|33.45(0.14)|32.81(0.07)|31.64(0.11)|30.83(0.06)|
>
> **Q6: How are the competing algorithms being shown fairly since they are not provided with these informative predicates? Could they be included as part of the observation space and run again?**
>
> A6: Thank you for your recommendation. It is important to note that the effectiveness of informative predicates is not solely reliant on their addition to the model; the relationships between these predicates (relations) are more crucial. Our plan follows the logic tree structure, and simply augmenting the state space with additional predicates without considering their dynamic interrelations would not be beneficial. We emphasize estimating dynamics that underpin these relations, which is a critical aspect of our framework. This approach ensures that our model not only incorporates informative predicates but also captures the essential relational dynamics for accurate predictions.
>
> **Q7:Minor Typos.**
>
> A7: These typos have been modified in a revision.

---

### Official Review · Reviewer_wzfu · 2023-11-01

**Soundness:** 3 good
**Presentation:** 3 good
**Contribution:** 3 good
**Rating:** 6
**Confidence:** 3

**Summary:**

Authors introduce a method that learns a logic tree and a policy using inverse reinforcement learning from observational data from experts, which is ultimately presented as a set of logical rules. Further, this logic tree allows the recovery of the reward function, using q-function estimation, which is then used for the reward estimation. State-action demonstrations are used as the raw input of the method. The logic tree is then generated in a top-down manner, through a transformer based model, which is then fed to a GAN model to estimate the agent policy. Evaluation was done both on synthetic (3 datasets) and real-world (1) datasets. Authors evaluate the method in a healthcare context and use MIMIC-3 and MIMIC-4 datasets. The F1 score and recall is measured for the predictions of the diagnosis, across 6 baseline models.

**Strengths:**

-Extracting logic rules in high-stakes domains is valuable and can facilitate better decision making.
-A good selection of baseline methods are compared, showing the strength of the proposed model.

**Weaknesses:**

There are several weaknesses of the method, detailed below.

-The major weakness I see in the paper is in the framing of the problem as an interpretability problem, where authors acknowledge the lack of interpretability in black box policies and how in high-stakes domains. The logic rules are presented as the high level explanations that can provide interpretability. While this is a reasonable assumption, this is not detailed further in the paper, or discussed in the evaluation section. E.g. A large logic tree might not be inherently interpretable, where the selection of the rules to be explained can be important.
- While there is a good selection of baseline methods compared, the synthetic/toy datasets need further additions of planning domains/datasets.

---- after the rebuttal ----

I have adjusted the scores accordingly after considering the rebuttal, and further considering the rebuttals and concerns of other reviewers.

**Questions:**

-What are the computation times for the MIMIC and synthetic datasets?  It will be helpful for the reader to understand the computation requirements.

---

> ### Author Response · Authors · 2023-11-21
> **Rebuttal by Authors**
>
> **Q1:The major weakness I see in the paper is in the framing of the problem as an interpretability problem, where authors acknowledge the lack of interpretability in black box policies and how in high-stakes domains.**
>
> A1: Thank you for your valuable feedback. In our paper, we constrain the size of the logic tree to enhance interpretability. Each branch within the tree uses a conjunction ("and") to maintain simplicity and directness in the reasoning process. Disjunctions ("or") are used between separate trees, indicating alternative reasoning paths to the root node. This structured approach ensures that only when all predicates in a path are true, a conclusion at the root is reached. Each distinct tree represents a unique reasoning pathway, simplifying the interpretability of complex decisions.
>
> **Q2: While there is a good selection of baseline methods compared, the synthetic/toy datasets need further additions of planning domains/datasets.**
>
> A2: Thanks for your suggestions. To demonstrate the performance of our framework, we evaluate our methods in benchmark tasks (the shifting maze [1]) for planning datasets.  This task involves a 2D point mass navigating to a goal position in a small maze when the position of the walls is changed between train and test time. At test time, the agent cannot simply mimic the actions learned during training, and instead must successfully infer that the goal in the maze is to reach the target. In this task, a reward is learned via IRL in the training environment, and the reward is used to reoptimize a new policy on a test environment. We compare our method with several competing IRL algorithms, including NLM, MemNN, MaxEnt-IRL, Deep PQR and AIRL. The results are shown in the following table. Our method still obtains the best performance.
>
> Table1: Results on planning dataset (the shifting maze).
> |  Methods   | NLM  | MemNN | MaxEnt-IRL | Deep PQR | AIRL | Ours |
> | :---  | ---:  | :--: | :--: | :--: | :--: | :--: |
> |  Reward  | 393.5 |372.8|353.8| 440.6| 384.4| 468.3|
>
>
>
> [1] Fu J, Luo K, Levine S. Learning robust rewards with adversarial inverse reinforcement learning[J]. arXiv preprint arXiv:1710.11248, 2017.
>
> **Q3: What are the computation times for the MIMIC and synthetic datasets? It will be helpful for the reader to understand the computation requirements.**
>
> A3: Following the reviewer’s suggestion, we show the runtime for all methods averaged on BlocksWorld tasks. Note that all methods are tested on an RTX 2080Ti GPU. The following table shows the time of 100 evaluation episodes. Our method is simpler to train at the same time achieving even better performance.
>
>
> Table2: Comparison of runtime.
> |  Methods   | NLM  | MemNN | MaxEnt-IRL | Deep PQR | AIRL | Ours |
> | :---  | ---:  | :--: | :--: | :--: | :--: | :--: |
> |  Runtime| 66.9s |101.3s|32.6s| 161.0s | 70.8s | 57.2s|

---

> > ### Comment · Reviewer_wzfu · 2023-11-22
> > **Rebuttal response**
> >
> > I thank the authors for providing additional results shifting domain and computation times. I have adjusted the scores accordingly after considering the rebuttal, and further considering the rebuttals and concerns of other reviewers.
> >
> > For the updated manuscript, please add the commentary around interpretability.

---

> ### Author Response · Authors · 2023-11-22
> **Invitation for further discussion**
>
> Dear Reviewer,
>
> I hope this message finds you well. Following your valuable feedback, we have diligently made revisions based on your previous feedback and are now eager to understand if there are any aspects that may still require refinement or additional clarification.
>
> Your insights are very important to us, and we eagerly await your response.
>
> Thank you for your time and effort.
>
> Best regards,
>
> The authors

---

### Meta-Review · Area_Chair_Ecjs · 2023-12-03

**Metareview:**

This paper proposes to deploy an IRL approach to extract logic trees from expert demonstrations.

On the positive side, all reviewers agree on the importance and soundness of the approach. Extracting logic trees could better assist decision making in especially high-stakes domains.  The main concern, shared by most reviewers during the discussion, is the relative lack of studies on interpretability and quality of the reward functions. While the authors provided some results during rebuttal by comparing a single baseline using a single benchmark for evaluating reward discovery, the reviewers still have questions on why the choice of only one benchmark and think a more comprehensive evaluations on this front would make the paper stronger.  Another point being discussed is the strong requirement of human-provided predicates that might have impacts on the practical performance.

**Justification For Why Not Higher Score:**

Reviewers, even the positive ones, have expressed reservations during the discussion due to the lack of studies on interpretability and quality of the reward functions.

**Justification For Why Not Lower Score:**

N/A

---

### Decision · Program_Chairs · 2024-01-16

Reject